



# 1  Key drivers of the annual carbon budget of biocrusts from various
# 2  climatic zones determined with a mechanistic data-driven model

Yunyao Ma[1], Bettina Weber[2,3], Alexandra Kratz[3], José Raggio[4], Claudia Colesie[5], Maik Veste[6,7],
Maaike Y. Bader[8], and Philipp Porada[1]
[1]Institute of Plant Science and Microbiology, Universität Hamburg, 22609 Hamburg, Germany
[2]Institute of Plant Sciences, Department of Biology, University of Graz, 8010 Graz, Austria
[3]Department of Multiphase Chemistry, Max Planck Institute for Chemistry, 55128 Mainz, Germany
[4]Farmacología, Farmacognosia y Botánica Department, Complutense University of Madrid, 28040 Madrid, Spain
[5]School of Geosciences, University of Edinburgh, EH9 3FF Edinburgh, United Kingdom
[6]Institute of Environmental Sciences, Brandenburg University of Technology Cottbus-Senftenberg, 03046 Cottbus, Germany
[7]CEBra - Centre for Energy Technology Brandenburg e.V., 03042 Cottbus, Germany
[8]Faculty of Geography, University of Marburg, 35032 Marburg, Germany
*Correspondence to*: Yunyao Ma (yunyao.ma@uni-hamburg.de)
**Abstract.** Biocrusts are a worldwide phenomenon, contributing substantially to ecosystem functioning. Their growth and
survival depend on multiple environmental factors, including climatic conditions. While the physiological responses of
biocrusts to individual environmental factors have been examined in laboratory experiments, the relative importance of these
factors along climatic gradients is largely unknown. Moreover, it is not fully understood how acclimation of biocrusts may
alter the relative impacts of certain factors. We aim here at determining the relative effects of environmental factors on
biocrusts along climatic gradients, using the carbon balance of biocrust organisms as a measure of their performance.
Additionally, we explore the role that seasonal acclimation plays in the carbon balance of biocrusts. We applied a data-
driven mechanistic model at six study sites along a climatic gradient to simulate the annual carbon balance of biocrusts
dominated by different lichen and moss species. Furthermore, we performed several sensitivity analyses to investigate the
relative importance of driving factors, thereby including the impacts of acclimation. Our modelling approach suggests
substantial effects of light intensity and relative humidity in temperate regions, while air temperature has the strongest
impact at alpine sites. In drylands, ambient $CO_2$ concentration and also the amount of rainfall are important drivers of the
carbon balance of biocrusts. Seasonal acclimation is a key feature, mostly in temperate regions, affecting biocrust
functioning. We conclude that climate change, which may lead to warmer and, in some regions, drier air, will potentially
have large effects on long-term carbon balances of biocrusts at global scale. Moreover, we highlight the key role of seasonal



acclimation, which suggests that the season and timing of collecting and monitoring biocrusts should be given additional
consideration in experimental investigations, especially when measurements are used as the basis for quantitative estimates
and forecasts.

## 1 Introduction

Non-vascular photoautotrophs, such as lichens, mosses, eukaryotic algae and cyanobacteria, together with heterotrophic
microorganisms form biological soil crusts (biocrusts) (Belnap et al., 2016). Biocrusts occur in various environments across
the globe and are especially ubiquitous in environments where low water availability inhibits the development of vascular
plants (Belnap et al., 2004; Lange et al., 1998a; Samolov et al., 2020). They provide a wide range of important ecosystem
functions, such as build-up of soil organic carbon and nutrients (Chamizo et al., 2012; Dümig et al., 2014; Ferrenberg et al.,
2018). Biocrusts contribute substantially to ecosystem carbon fluxes at the global scale (Elbert et al., 2012). Especially in
desert ecosystems, biocrusts can be a major contributor to the annual $CO_2$ uptake (Tucker et al., 2019; Wohlfahrt et al., 2008).
Cyanobacteria, which are common components of biocrusts, either free-living or associated with mosses and lichens,
facilitate biotic nitrogen fixation and may subsequently increase carbon sequestration by enhancing the concentration of
photosynthesis-related enzymes in top soils (Ackermann et al., 2012; Evans and Lange, 2003).
Owing to their importance in ecosystem functioning, studies on growth and survival of biocrusts are crucial. Growth depends
on the long-term carbon balance (hereafter, C balance), which corresponds to the (accumulated) net carbon flux across the
system boundaries including all relevant carbon gains and losses. In order to ensure survival, any species needs to achieve a
positive C balance in the long-term, while negative values may occur for short periods. Acknowledging the importance of C
balance, an increasing number of studies have investigated the long-term C balance of individual non-vascular organisms as
well as biocrust communities, and their environmental drivers. An annual carbon budget based on measured field data was
reported in the study of Lange (2003b) on the crustose lichen *Lecanora muralis* growing on a rock surface in a temperate
climate of southern Germany. Furthermore, Büdel et al. (2018) measured the annual C balance for a cyanobacteria-
dominated biocrust in an Australian dry savannah ecosystem. Several other studies obtained long-term, large-scale values of
the C balance by scaling up short-term, local measurements of $CO_2$ exchange rate under natural field conditions (Lange et al.,
1994; Zotz et al., 2003). For an estimation of the global C balance of cryptogamic covers, which include biocrusts,
conversion factors based on the maximum photosynthesis rate have been suggested as a best-guess solution (Elbert et al.,
2012). However, there are some drawbacks to these approaches for acquiring a C balance at both organism and community
scale. First, the measurement of the long-term continuous $CO_2$ exchange rate of an individual organism or biocrust
community has technical limitations and is highly time- and resource-consuming. Second, upscaling via extrapolation may
result in bias in annual C balance estimation if the length and the frequency of sampling cannot capture the full variability of
$CO_2$ exchange throughout the year (Bader et al., 2010). Moreover, using empirical approaches alone, it is difficult to



understand the underlying mechanisms of how climatic conditions affect individual physiological processes, and
consequently which role these processes play in the observed changes in C balance and growth at the individual as well as
community level. Such approaches are thus subject to large uncertainty when used for projections of C balance under climate
change.
Most studies on the relationships between C balance and environmental factors for biocrusts are based on laboratory
experiments (e.g. Coe et al., 2012; Cowan et al., 1992; Lange et al., 1998a) or direct field measurements in situ over short
periods of time (e.g. Brostoff et al., 2005; Lange et al., 1994). From this work cited above, it has been recognized that the
type and temporal pattern of water supply, temperature, radiation, and also $CO_2$ concentration are among the factors that are
crucial for the C balance of biocrusts. While highest values of productivity in biocrusts are achieved when the environmental
factors are in their optimal range, it has been found that biocrusts under field conditions are also able to achieve maximum
activity and thus, potential productivity, under sub-optimal conditions of temperature and light (Colesie et al., 2016; Raggio
et al., 2017, 2014). It is largely unknown, however, which relative importance each of these factors has for the long-term C
balance of biocrusts under natural field conditions, and if the importance of factors shows a spatial and temporal pattern. In
addition, seasonal acclimation of photosynthetic and respiratory properties of species to intra-annually varying climate
factors found by several studies (e.g. Gauslaa et al., 2006; Lange and Green, 2005; Wagner et al., 2014) may substantially
affect biocrust C balance, thus leading to further complexity in the spatio-temporal relations between C balance and
environmental factors. One of the few experimental studies investigating biocrust acclimation potential to changing
temperatures has found threshold temperatures for the survival of lichen species (Colesie et al., 2018) but the overall extent
of the impact is poorly understood.
Here, we applied a mechanistic data-driven model to (a) complement empirical estimates of the annual C balance of
biocrusts and (b) to address the knowledge gaps concerning the relative importance of different environmental factors for the
C balance along climatic gradients, thereby accounting for the role of seasonal acclimation. The advantage of this modelling
approach is that it can predict at high temporal resolution the dynamic C balance of biocrust organisms for given locations by
simulating the physiological processes driven by environmental factors. The model allows for a deeper mechanistic
understanding of the C balance of biocrusts through factorial experiments and sensitivity analyses regarding parameters and
individual environmental factors, which would be impractical to realize in field or laboratory experiments. To complement
our analyses using the data-driven model, a process-based dynamic non-vascular vegetation model, called LiBry, was
employed as an supporting tool (Porada et al., 2013).
**2 Material and Methods**
We simulated the C balance of biocrusts from six climatically different study sites in a semi-empirical way using a data-
driven model. The model simulates photosynthetic rate based on the Farquhar photosynthesis model developed by Farquhar
and von Caemmerer (1982) and respiration rate based on Q10 relationship. The C balance is computed as the difference of



photosynthesis and respiration accumulated over a given time period. In the model, both photosynthesis and respiration
depend on surface temperature, relative water saturation, and the activity of the biocrust, which are all simulated in a coupled
way via the surface energy balance as a function of climate input data. Photosynthesis additionally depends on ambient $CO_2$
concentration.
To calibrate the model, we first determined soil and land surface properties that are required for the coupled energy and
water balance in the model through fitting simulated to measured surface temperature patterns. Then, we parameterized the
physiological properties of the organisms using measured relationships between net photosynthesis and light intensity, water
content, and temperature. Finally, we validated the model with regard to the water content or activity patterns of biocrusts
and compared the modelled $CO_2$ assimilation rate to measured values. The data sets used for calibration and validation of the
model as well as basic climate conditions of each site are described in Table 1. Sites were listed in ascending order of total
annual precipitation based on measured data.

**2.1 Study sites**

In our study we considered six sites, namely two dryland sites at Almeria (Spain) and Soebatsfontein (South Africa;
hereafter D1 and D2); three temperate sites at Gössenheim (Germany), Öland (Sweden), and Linde (Germany; Hereafter T1,
T2, and T3); and one alpine site at Hochtor (Austria; Hereafter A1) (Table 1). These sites were chosen based on data
availability and because they cover a broad range of different climatic conditions. To our knowledge, the necessary empirical
data regarding climatic conditions, species physiological characteristics, and status such as activity, which is used to estimate
C balance, have been monitored in only a small number of experiments, including the six study sites chosen here.
Sites D1 and D2 are characterized by a semi-arid climate with mean annual precipitation of less than 250 mm, but a wet
winter season (Büdel et al., 2014; Haarmeyer et al., 2010). Sites T1, T2, and T3 have a temperate climate. The mean
precipitation in these three sites is around 550 mm (Büdel et al., 2014; Diez et al., 2019). Site A1 is located in a humid alpine
region with a mean annual precipitation between 1750 and 2000 mm, of which more than 70% are snowfall; the A1 site is
covered by snow for at least 200 days per year (Büdel et al., 2014). More detailed site descriptions are provided in the
corresponding studies cited above.

**2.2 Observational Data**

**2.2.1 Climatic variables**

The proposed data-driven model for estimating the annual C balance of dominant biocrust types at each site was forced by
hourly microclimatic variables. The forcing data sets of the data-driven model include photosynthetically active radiation
(PAR), long-wave radiation (near-infrared), relative air humidity, air temperature, wind speed, rainfall, and snowfall. All the
microclimatic variables were measured on-site by climate stations with a temporal resolution of 10 min (5 min in A1 and D1;
data available at http://www.biota-africa.org; Raggio et al., 2017; M. Veste, unpublished data), except for long-wave





radiation and snowfall, which were taken from ERA5 dataset (https://www.ecmwf.int/en/forecasts/datasets/reanalysis-
datasets/era5). Then all these microclimate data were aggregated to data with hourly temporal resolution.

### 2.2.2 Dynamic biocrust variables

Surface temperature data are available for all sites. Biocrust activity was either monitored directly (binary: active or not
active) using a continuous chlorophyll fluorescence monitoring system (Raggio et al., 2014, 2017), or indirectly via the
electrical conductivity of the substrate (BWP, Umweltanalytische Produkte GmbH, Cottbus, Germany; Weber et al., 2016;
M. Veste, unpublished data). For site D2, the biocrust water content was calculated instead of activity based on electrical
conductivity. Due to snow covering the measuring instruments, data of site A1 only covers the time from August to October.
Samples from both lichen- and also moss-dominated biocrusts were measured at all sites, except for site T3 where four
BWPs were mostly located in moss-dominated biocrusts. At site D2, additionally cyanolichen- and cyanobacteria-dominated
biocrusts were monitored. The measured surface temperature, water content, and activity data at all sites were then
aggregated to data with a temporal resolution of one hour.
We did not directly use the observed surface temperature and activity as forcing data for the model since these properties are
strongly linked to water saturation (and $CO_2$ diffusivity). Input data of water saturation, however, were not available at most
sites. Although the overall patterns of simulated and observed surface temperature match well (see below, Sect. 2.3),
inconsistencies would likely occur at hourly resolution if simulated dynamic water content was used in the model together
with observed temperature and activity. Hence, the time-series of surface temperature and water saturation data at all sites
were estimated based on a simulation of the energy and water balance. The activity of the organisms was then approximated
via the empirical equations describing the link between water saturation and metabolic activity (see Porada et al., 2013).
Furthermore, ambient $CO_2$ concentration was assumed to be constant at 400 ppm.
For validation of C balance, we used data of the on-site $CO_2$-exchange rate of different biocrust types (lichen- and moss- and
also cyano-dominated biocrusts; the latter composed of cyanolichen and cyanobacteria) that were measured by a portable
gas exchange system at several time intervals from November 4[th] to 8[th] at site D2 (Tamm et al., 2018). For the other sites,
additional field measurements of $CO_2$-exchange were not available.

### 2.2.3 Photosynthesis response and water storage

For all sites, $CO_2$ exchange measurements under controlled conditions in the laboratory or in the field (site T3) were
conducted using a mobile gas exchange system GFS 3000 (Walz GmbH, Effeltrich, Germany) with an infrared-gas analyzer
to explore the physiological characteristics of samples of different biocrust types (same as those measured for validation;
main species see Table 1; Diez et al., 2019; Raggio et al., 2018; Tamm et al., 2018). Net photosynthesis was measured at
different ranges of light intensity, water content, and temperature. Light response curves, for instance, were determined at
optimum water saturation and 15 °C, water response curves were measured at 400 µmol m$^{-2}$ s$^{-1}$ and 15 °C at sites D1, T1, T2,
and A1 (Raggio et al., 2018). Moreover, the maximum water storage capacity (MWC) of the samples was quantified in the





laboratory for samples from sites D1, T1, T2, and A1 (Raggio et al., 2018), whereas the MWC at site D2 was approximated
as the maximum value when measuring water response curves (Tamm et al., 2018; Weber et al., 2012). MWC at site T3 was
estimated as the value of the same genus measured in Hamburg, Germany (*Cladonia portentosa* and *Polytrichum formosum*,
Petersen et al., in prep.). MWC was acquired since it is one of the essential parameters of the model to convert the specific
water content in mm to relative water saturation required by the model used here.
**Table 1**: Properties of the study sites and data which are available (+ sign) for calibration and validation of the data-driven
model

| Site | | Almeria, Spain | Soebatsfontein, South Africa | Gössenheim, Bavaria, Germany | Öland, Sweden | Linde, Brandenburg, Germany | Hochtor, Austria |
|---|---|---|---|---|---|---|---|
| Code | | D1 | D2 | T1 | T2 | T3 | A1 |
| Climate | | semi-arid | semi-arid | Temperate | Temperate | Temperate | Alpine |
| Measured annual rainfall [mm] | | 110 | 141 | 424 | 441 | 449 | 744 |
| Dominant species at the site | | *Psora decipiens, Didymodon rigidulus* | *Psora decipiens, Psora crenata, Ceratodon purpureus, Collema coccophorum* | *Psora decipiens, Trichostomun crispulum* | *Psora decipiens, Tortella tortuosa* | *Cladonia furcata, Polytrichum piliferum* | *Psora decipiens, Tortella rigens* |
| Data for Calibration | Laboratory $CO_2$ exchange response curves | Light, water, temperature | Light, water, temperature | Light, water, temperature | Light, water, temperature | Light, temperature | Light, water, temperature |
| | Surface temperature | + | + | + | + | + | + |
| Data for Validation | Water content | - | + | - | - | - | - |
| | Activity | + | - | + | + | + | + |
| | $CO_2$ exchange on site | - | + | - | - | - | - |





| | | | | | | |
|---|---|---|---|---|---|---|
| References | Raggio et al., 2018 | Tamm et al., 2018; Weber et al., 2012 | Raggio et al., 2018 | Raggio et al., 2018 | Veste, unpublished data; Diez et al. 2019 | Raggio et al., 2018 |

## 2.3 Parameterization of the data-driven model

### 2.3.1 Abiotic surface properties

Several abiotic parameters of the data-driven model describing soil and land surface properties, such as roughness length or soil thermal conductivity, were required to simulate the energy and water balance. These parameters were obtained by fitting the daily and diurnal surface temperature patterns of lichen-dominated biocrust at all sites except for site T3. At site T3, we compared the surface temperature patterns of simulated moss-dominated biocrusts to data collected by sensors in four locations.

The set of parameters that corresponded to minimum differences between simulated and measured values (visual assessment) was used in the data-driven model. The calibration results of surface temperature and the photosynthesis response curves at site T2 are shown in Fig. 1 and Fig. 2, respectively. The results of dominant species at other sites are shown in Fig. S1 and S2.

The daily surface temperature was simulated accurately (visual comparison) except for site T3 where the temperature during cold seasons was underestimated, and at site D1 the peak temperature within a day in hot seasons was underestimated (Fig. S1). The peak in surface temperature occurred too early by around 3 hours at site T1 and T2, but the magnitude of the peak corresponded well to the measured data (Fig. 1 and S1). Therefore, in general, the fitting of the surface temperature patterns was satisfactory.

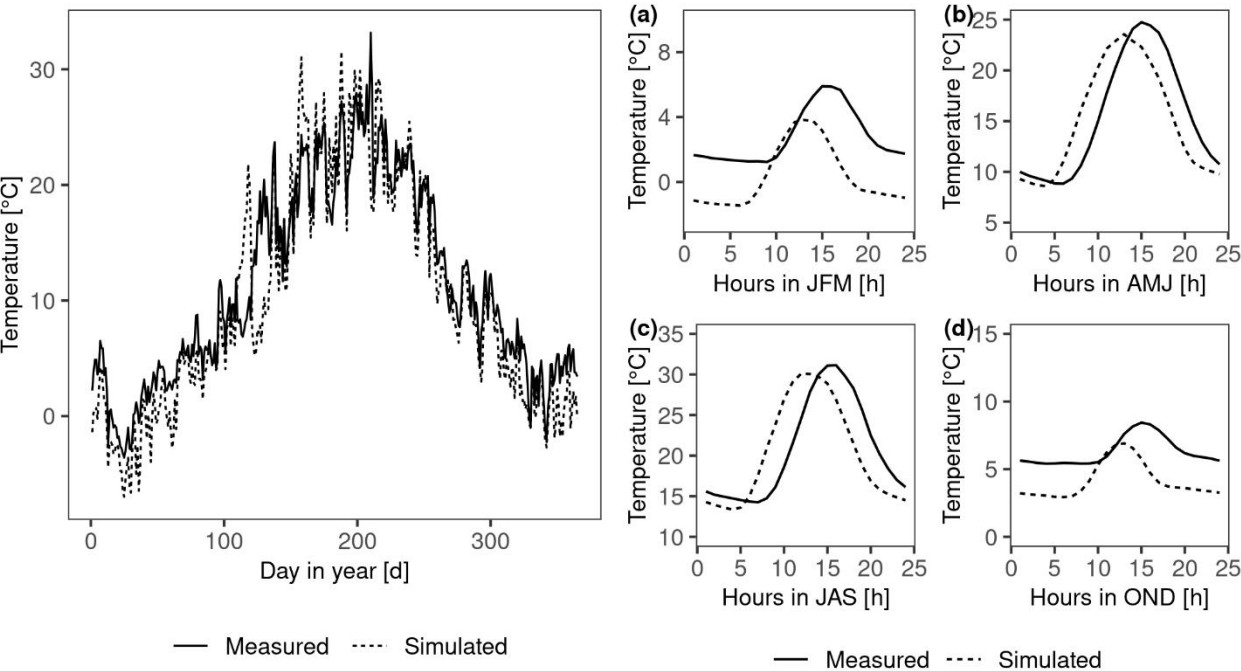

179

**Figure 1**: Calibration results of abiotic parameters of the data-driven model by fitting the daily (left panel) and diurnal (right panel) patterns of surface temperature at site T2. Right: (a) to (d) represent the patterns of average hourly surface temperature from January to March (JFM), April to June (AMJ), July to September (JAS), and October to December (OND), respectively.

### 2.3.2 Biocrust physiological properties

Furthermore, several parameters required by the Farquhar photosynthesis scheme and the respiration scheme were directly measured or calculated from gas exchange data sets, such as MWC of the thallus, optimum water content, the optimum temperature for gross photosynthesis, reference maintenance respiration rate, and the $Q_{10}$ value of respiration. Since the temperature range at all sites except D2 was too small to capture the optimum temperature reliably, it was approximated from the measured data set as the average surface temperature during active periods. In addition, the optimum temperature was also constrained by fitting the Farquhar equations to photosynthesis curves, as related to environmental factors light, water content, and temperature. Such fitting method was also used to obtain some other photosynthesis-related parameters of organisms, such as molar carboxylation and oxygenation rate of RuBisCO (Vcmax, Vomax), respiration cost of RuBisCO enzyme (p_rr), and water saturation at which organisms become active (satmin).





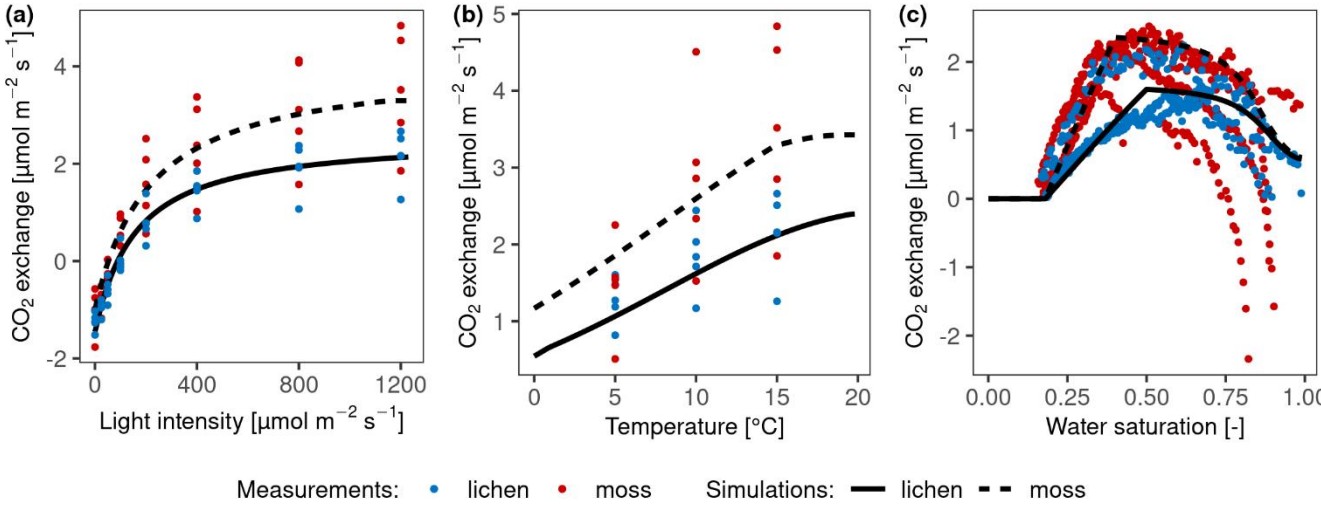

194

**Figure 2**: Calibration of photosynthesis parameters of the model by fitting photosynthesis response curves of moss- and lichen-dominated biocrust samples to measurements at site T2. (a): net photosynthesis rate in response to light at optimum water content and 15 °C. (b): net photosynthesis rate in response to temperature at 1200 µmol m$^{-2}$ s$^{-1}$ light and optimal water content. (c): net photosynthesis rate in response to relative water saturation at 400 µmol m$^{-2}$ s$^{-1}$ light and 15 °C.

Since the measured values between replicates showed large variation, in particular the water and temperature-dependent data, as illustrated by the coloured points in Fig. 2, we fitted the curves to the average values of all replicates. The calibration results showed that visually and overall, the photosynthetic curves could be parameterized to fit the measurements well, given that different samples were used for measuring responses to different driving factors, and considering the methodological differences between light and water response curve measurements. However, the water responses were least well-fitted, especially at high water contents, because the measured photosynthesis response rate can become negative at high water saturation, but it is impossible to simulate negative photosynthesis rates with the Farquhar photosynthetic model for the light and $CO_2$ conditions of the laboratory setup. Under these conditions, photosynthesis is larger than respiration. Thus, even under low diffusivity, caused by high water saturation, there will be no net diffusion of $CO_2$ from the thallus into the ambient air. Furthermore, matching the simulated temperature-response of net photosynthesis to the measured data under cool conditions turned out to be difficult for samples from site T1 and A1 (Fig. S2). There were too few data points in the measured temperature response data set to constrain the optimum temperature and temperature relation (see Fig. 2), but the fitting turned out well because the simulated optimum temperature for net photosynthesis at site T1 was coincidentally close to the value of 17 °C reported by Colesie et al. for this site. (2014; see Fig. S2).

**2.4 Validation of the data-driven model**



The water saturation and activity estimated by the data-driven model were validated by comparing with the daily and diurnal
patterns of measured activity (5 sites, not at site D2) and water content data (only at D2). Furthermore, the C balance
estimated by the data-driven model was validated by comparison to the in situ measured net $CO_2$ exchange rate of moss,
lichen, and cyanocrust-dominated biocrusts. These data were sampled at site D2 by removing the soil respiration rate,
predicted by means of a fitted regression (Weber et al., 2012). Since data on water saturation were available, measured PAR,
surface temperature, and water content were used to simulate the C balance using the data-driven model, in contrast to the
setup described above. The activity, however, was calculated in the same way as described in the setup. Moreover, the
parameters of the model were the same as the calibrated ones of the corresponding biocrust types.
**2.5 Sensitivity analysis**
**2.5.1 Effects of environmental factors**
To investigate the role that environmental factors, namely air temperature (Tair), light intensity (Light), ambient $CO_2$
concentration (CO2), and different types of water sources play in regulating the C balance of biocrusts, sensitivity analyses
were conducted using our data-driven model for lichen-dominated biocrusts from all study sites. The different types of water
sources include rainfall (Rain) and non-rainfall water inputs such as dew and water vapor, which are also determined by
relative air humidity (Rhum).
All the environmental factors were reduced and increased by half (+/- 50%), except for air temperature and relative humidity.
The air temperature differences varied by 5 K and relative humidity by 20%. Moreover, relative humidity was constrained
between 0 and 100% when the varied relative humidity exceeded this range. The annual C balance with changing
environmental factors was then normalized for comparing the relative importance of factors among climate zones following
Eq. (1):
$Normalized\ C\ balance = \frac{C_{ij} - C_j}{|C_j|},$ \hfill (1)
where $C_{ij}$ is the C balance of factor j under operation i, and $C_j$ is the original C balance of factor j.
Then the positive normalized C balance would show an increased annual C balance with varying environmental factors, and
thus more carbon accumulation. Moreover, the size of the normalized C balance is proportional to the magnitude of change
of the C balance when certain environmental factors change. Therefore, a larger normalized C balance also demonstrates a
larger change in annual C balance, and thus a larger effect of this environmental factor.
To interpret the spatial distribution of the importance of different environmental factors on C balance, the relative
importance of each factor in the given climatic region was calculated following Eq. (2) and Eq. (3):
$N_{ab} = |N_{ab}(increase)| + |N_{ab}(decrease)|,$ \hfill (2)



$Relative\ importance = \frac{N_{ab}}{\sum_{a=1}^{n} N_{ab}}$,  (3)
where $N_{ab}$ (increase) and $N_{ab}$ (decrease) are the normalized C balance of increasing or decreasing the environmental factor a
in climatic region b, respectively. $N_{ab}$ is thus the change amplitude of normalized C balance of environmental factor a in
climatic region b.
**2.5.2 Effect of seasonal acclimation**
Another sensitivity analysis was performed for site T1 to investigate the impact of seasonally acclimatized properties on
carbon assimilation. The properties were varied based on the literature. Respiration of lichens was found to acclimate to
seasonal changes in temperature (Lange and Green, 2005). Moreover, under low light, organisms showed shade-adapted
physiological characteristics with low PAR compensation and saturation points (LCP and LSP; Green and Lange, 1991).
Thus, under low light conditions, the organisms have a stronger ability to utilize low light intensities for photosynthesis.
These properties can be expressed by certain parameters of the data-driven model. For instance, the respiration rate is
determined by the parameter metabolic respiration cost per surface area (Resp_main); LCP and LSP can be affected by
changing the slope of the photosynthesis-light relations through light absorption fraction in cells (extL); LCP and LSP can
also be modified via the ratio of Jmax to Vcmax (jvratio) as it influences the value of light use efficiency at unsaturated light.
With higher efficiency, the light required to reach the saturated light level declines. Jmax is a crucial parameter quantifying
the maximum rate of electron transport in the light-dependent reactions of photosynthesis, Vcmax describes the maximum
rate of carboxylation of RuBisCO in the Calvin Cycle of photosynthesis (Walker et al., 2014). Accordingly, rather than
keeping all parameters fixed throughout the simulation period of the data-driven model, in the sensitivity analysis, the
physiological parameters were set to another set of values in the winter months. We analyzed the lichen- and moss-
dominated biocrusts at site T1 as an example, because the measured time-series activity showed that in temperate sites such
as T1, the organisms were active most of the time, and thus the C balance would be more sensitive to the modifying
properties.
In the sensitivity analysis, these calibrated physiological parameters of the data-driven model were varied for the non-
growing months to adapt to the climatic conditions because the organisms at site T1 were collected in their growing seasons.
Specifically, in an hourly simulation during September and December, January, and February, the parameter Resp_main was
reduced to half to lighten the respiratory cost for the samples collected at site T1. The size of extL was doubled to increase
the slope of photosynthesis-light relations. In addition, the parameter jvratio was doubled as well to enhance the light use
efficiency.
**2.6 LiBry Model**





LiBry is a process-based dynamic global vegetation model (DGVM) specific to non-vascular vegetation. The model mimics
environmental filtering in the real world by simulating many different functional strategies and selecting those which
maintain a positive C balance under the respective climatic conditions. The strategies are characterized by a combination of
11 physiological and morphological parameters. More information about the model is briefly described in the Appendix, and
a full detailed description can be found in Porada et al. (2013, 2019). For this study, the LiBry model was run for 300 years
with repeated microclimate forcing data of one year from the six study sites, calibrated abiotic parameters same as the data-
driven model, and initially generated 1000 strategies. C balance and dynamics of the surface cover of the strategies were
simulated until a steady state was reached, so that the final successful strategies were those where long-term biomass values
were positive. Moreover, at the end of the simulation, the average values of functional traits were estimated by weighting all
surviving strategies based on their relative cover. The (hypothetical) strategy characterized by these average values is called
average strategy. The strategy with the largest cover area is called dominant strategy.
Furthermore, we compared the physiological parameters of the average strategy and the selected dominant strategies to the
ones of organisms in the field by means of their respective photosynthesis response curves. This comparison can verify the C
balance estimated by the data-driven model from a reversed perspective as the strategies were freely selected by the LiBry
model based on their C balance, without prescribing values based on site level observations.
**3 Results**
**3.1 Data-driven model**
**3.1.1 Validation of the data-driven model**



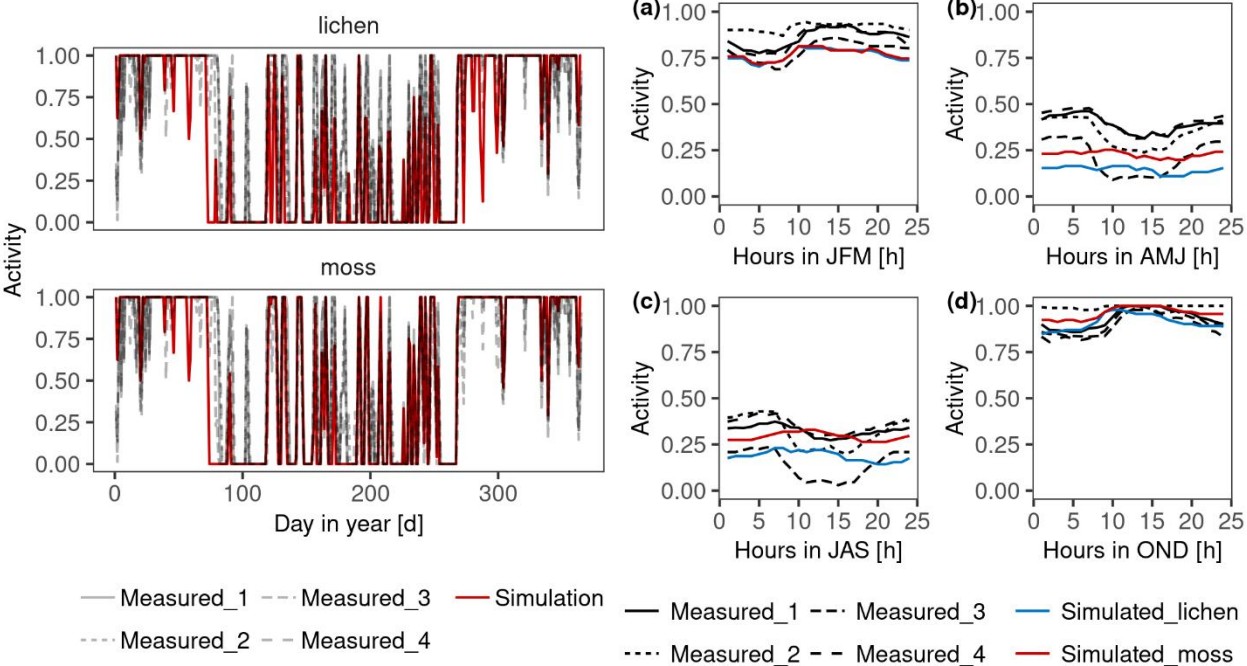


**Figure 3**: Validation of the estimated daily (left panel) and diurnal (right panel) patterns of activity of lichen- and moss-dominated biocrusts at site T3. The simulated patterns of activity were compared to measured data by four sensors at different locations. Right: (a) to (d) represent the patterns of average hourly activity from January to March (JFM), April to June (AMJ), July to September (JAS), and October to December (OND), respectively.

In general, the simulated daily and diurnal patterns of activity (water content at site D2) fit the measurements reasonably well in magnitude (Fig. 3 and Fig. S3). However, our fitting resulted in a more dampened diurnal activity pattern simulated by the model, and the activity at night and in the morning was underestimated during several seasons at sites D1, T1, T2 and A1. In addition, both the daily and diurnal activity during April and June at site T1 were underestimated. Furthermore, water content was overestimated for moss-dominated biocrust, especially when there was a large amount of water input at D2, although the patterns corresponded well to the measured data for all three biocrust types (Fig. S3). This overestimation may have resulted from the bias in measured MWC of samples used for constraining the water content in the model.

The comparison of simulated and on-site measured $CO_2$ exchange rates of three biocrust types (moss, lichen, and cyanocrust composed of cyanolichens and cyanobacteria) at site D2 showed mismatches, especially when water saturation was at both ends of the gradient (Fig. 4). The $CO_2$ exchange rate at high water content was overestimated compared to the measurements. Moreover, there were large variations in measurements of respiration and $CO_2$ exchange rate as water content was low and thus simulated $CO_2$ exchange rate was zero. Excluding the values at both ends of water content (0.58 and 1.74 mm for moss; 0.22 and 0.68 mm for lichen; 0.26 and 0.65 mm for cyanocrust), the accuracy of the model predictions was improved (root mean square error (rmse) decreased from 1.45 to 1.36 for moss, 1.27 to 0.71 for lichen, and 0.92 to 0.87 for cyanocrust).



Furthermore, the simulations were similar to measurements in magnitude. Therefore, despite the large variation, we are
confident about the general validity of the model.

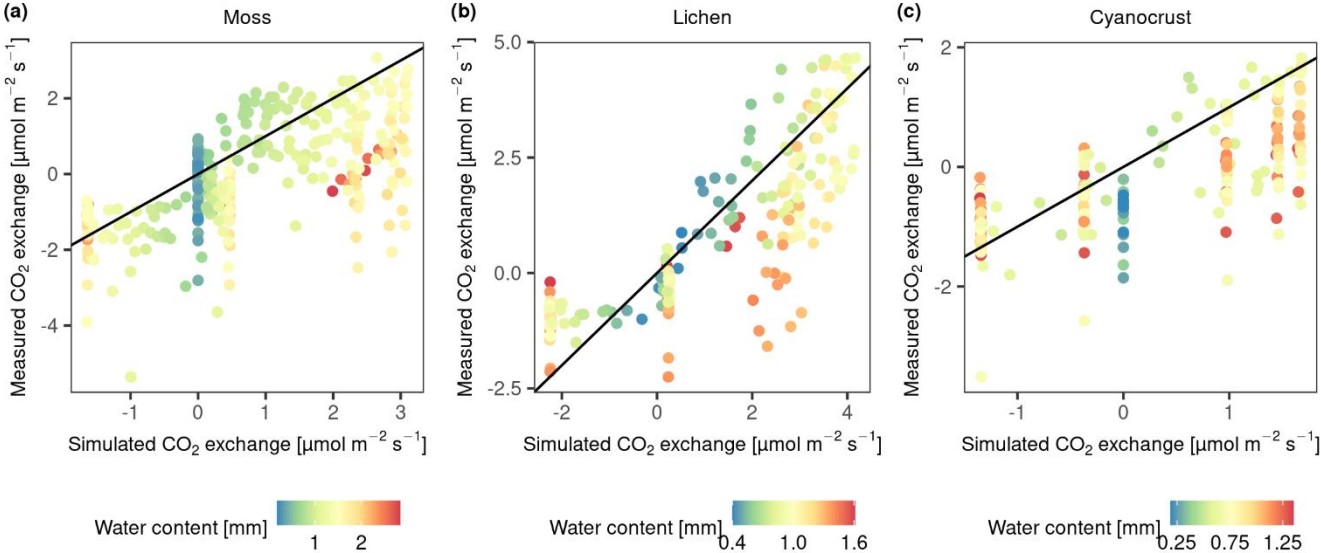


**Figure 4**: Validation of the photosynthesis and respiration scheme of the data-driven model through comparison of modelled
and measured $CO_2$ exchange rate of moss, lichen, and cyanocrust given the measured water content, surface temperature,
PAR, and calculated activity. Observational data were collected in November in 2013 at site D2. The 1:1 line is shown in
black.

**3.1.2 Estimated C balance by data-driven model**

The simulated annual C balance of each collected biocrust type at each site is listed in Table. 2. The annual C balance of
lichen- and moss-dominated biocrusts at two dryland sites showed a small positive value. Moreover, a cyanocrust
additionally measured at site D2 showed a small net release of carbon in the model.
**Table 2**: Simulated annual carbon budgets of each biocrust type at all sites

|  | Lichen | Moss | Cyanocrust |
|---|---|---|---|
|  | g C m$^{-2}$ yr$^{-1}$ | g C m$^{-2}$ yr$^{-1}$ | g C m$^{-2}$ yr$^{-1}$ |
| D1 (Almeria) | 3.8 | 3.2 |  |
| D2 (Soebatsfontein) | 0.7 | 6.3 | -2.0 |
| T1 (Gössenheim) | -27.3 | -28.6 |  |

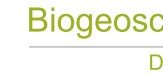
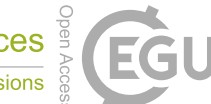

| | | |
|---|---|---|
| T2 (Öland) | -96.0 | -63.9 |
| T3 (Linde) | 7.1 | 13.3 |
| A1 (Hochtor) | -11.4 | 6.2 |

Furthermore, according to these data-driven model simulations, despite the C balance of two biocrust types being positive at
site T3, a large amount of carbon was lost at the sites T1 and T2 in temperate humid regions. These results imply that
according to the data-driven model, the biocrusts would not survive in the long-term at most of the temperate humid research
sites. At the alpine site A1, the moss-dominated biocrust showed a small positive C balance, whereas the lichen crust lost
carbon in a year with long periods of ice cover.
**3.2 Dominant strategies selected by the LiBry Model**
In general, the photosynthesis response curves of dominant and average strategies selected by the LiBry model did not fit
well to the measurements, especially at temperate site T2 (Fig. 5; the results for the other sites with negative C balance are
shown in Fig. S4). Specifically, the selected physiological traits which determine water and light acquisition of the dominant
and average strategies in LiBry differed from those of the collected samples at all sites. Compared to the measured samples,
the LiBry strategies showed markedly higher efficiency at low light intensity and faster activation. By design, the Libry
model selected strategies with a positive C balance in the long-term run, and thus the mismatches are consistent with the fact
that the data-driven model simulated negative C balances.

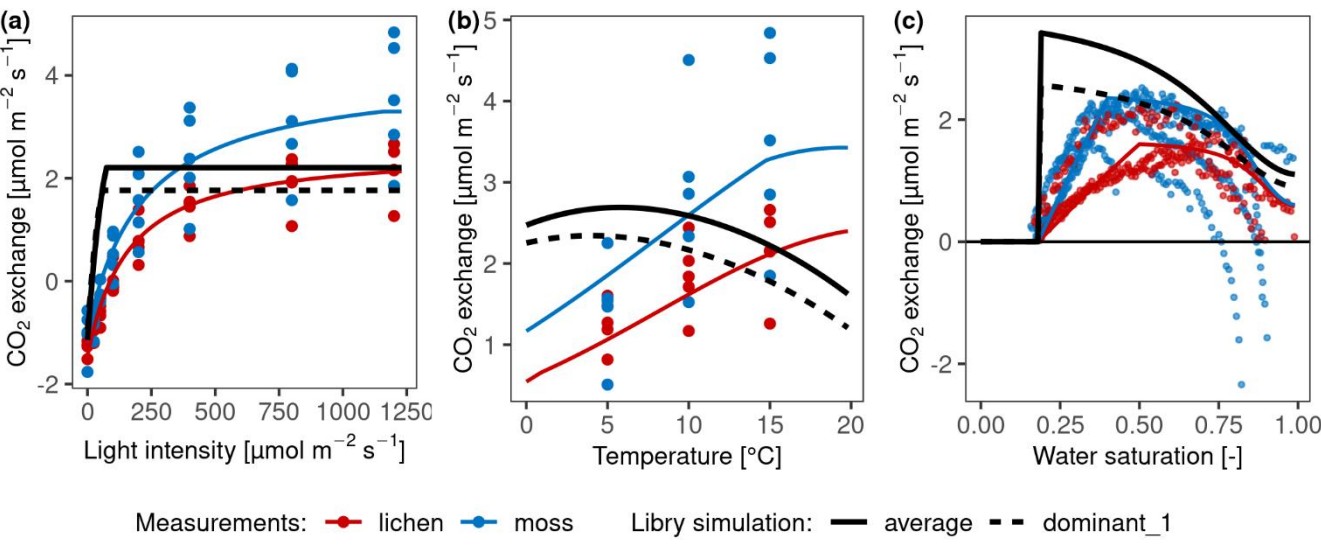


**Figure 5**: Comparison of net photosynthesis response of measured samples with simulated selected dominant and average
strategies of LiBry at site T2. (a): light response curve; (b): temperature response curve; (c): water response relation. The





colored points represent the measured $CO_2$ exchange rates of moss and lichen, and the coloured lines correspond to the data-driven model. The black lines show the photosynthesis response of the dominant strategy selected by the LiBry model (dashed) and the average strategy (solid). The parameter values of the average strategy correspond to the average of all surviving strategies.

**3.3 Driving factors of variation of the C balance**

**3.3.1 Environmental factors**

The environmental factors light intensity, $CO_2$ concentration, air temperature, and various water sources, had different effects on the C balance of lichens in different climate zones (Fig. 6). For all sites within a given climate zone, the effects of different environmental factors on C balance were overall similar but showed an apparent larger variation at the temperate site T3 in contrast to the other two temperate sites, and at site D2 compared to D1 (Fig. 6a). This may be due to physiological differences of the investigated biocrust species between these sites and consequently variations in the responses of net photosynthesis rate to temperature, water, and light between them (Fig. 2 and S2).

Furthermore, the spatial patterns of the relative importance of different environmental factors show that the factors which have the strongest effects differ between climatic regions (Fig. 6b).



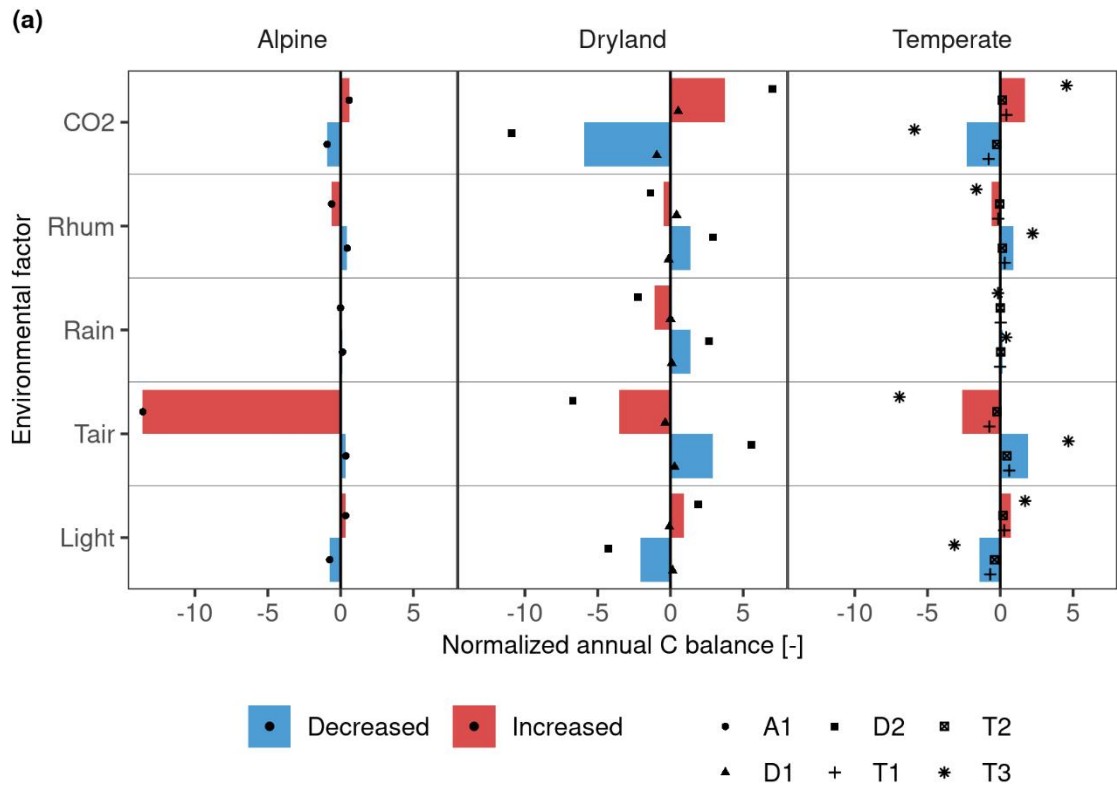

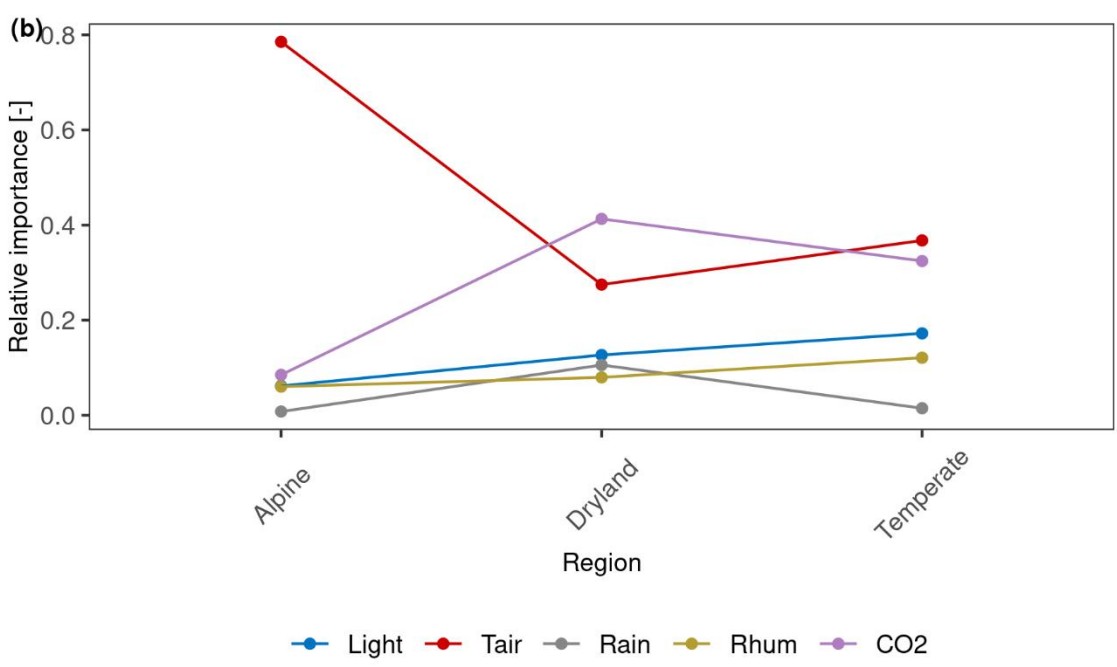




**Figure 6**: (a) The effects of environmental factors - $CO_2$ concentration (CO2), relative air humidity (Rhum), rainfall amount
(Rain), air temperature (Tair) and light intensity (Light) on the annual C balance of lichens in different climate regions. The
altered annual C balance resulting from increasing or decreasing environmental factors is normalized by the C balance under
original environmental conditions. The colored columns indicate the average value of the normalized C balance at sites with
similar climate conditions. Various styles of black points indicate different sites. Positive normalized C balance implies that
the annual C balance increases with varying environmental factors and more carbon was accumulated in a year at the site,
and vice versa. A larger normalized C balance reflects that the C balance is more sensitive to the altering environmental
factor, and thus the environmental factor has a larger effect on C balance. (b) Relative importance of each environmental
factor compared to other factors across the climatic regions. Larger relative importance implies a more important effect the
factor has on the C balance compared to other factors in the given climatic region, and vice versa.
In general, air temperature and $CO_2$ concentration were the most important drivers for C balance of biocrust organisms
between climate zones. Light intensity and relative humidity played a relevant role in impacting the C balance as well.
Rainfall amount had lower relative importance at all sites except dryland D2, where the effect of rainfall on C balance was
similar to other factors (Fig. 6a). Therefore, rainfall amount showed a maximum in relative importance in drylands,
compared to other regions. In general, the effect of the other water source, relative air humidity, was moderate but notable at
all climate zones, and is slightly larger in temperate region in comparison to other climate zones. Furthermore, the humidity
had a slightly larger impact on C balance in comparison to rainfall amount at all temperate and alpine sites (e.g., change
amplitude was 0.04 for rainfall and 0.44 for humidity at T1). Moreover, reduced humidity can have a positive effect on C
accumulation in these regions (e.g., the normalized C balance was 0.3 at T1 when relative humidity was reduced). In
drylands, however, the impacts of water sources on C balance varied between sites. The results showed that relative
humidity had a larger impact than rainfall amount at D1 while similar at D2. Reducing relative humidity had a positive effect
on C accumulation at D2 while C accumulation was reduced at D1 (normalized C balance is 2.90 at D2 and -0.13 at D1
when relative humidity decreases).
The ambient $CO_2$ concentration was an essential factor for the C balance at all sites especially in drylands, resulting in
positive effects on C balance with increasing $CO_2$. Furthermore, light intensity had a marked impact on the C budget at all
sites except for dryland D1, and it was relatively more important in temperate regions. At site T2, for example, the
normalized C balance was changed to -0.39 and to 0.19 for half and doubled light intensity, respectively. At these sites, the
normalized C balance increased with enhanced light intensity. At site D1, however, the values did not vary largely, and even
slightly more carbon was lost as the light levels increased (0.16 and -0.08 for half and doubled light intensity). Air
temperature had a large impact on C balance at all sites. Especially at alpine site A1, C balance decreased strongly as air
temperature raised by 5 K (normalized C balance of -13.59), and at all sites, the direction of the effect remained constant,
namely, warming decreased the C balance and vice versa.

### 3.3.2 Acclimation of physiological properties

The sensitivity analysis for acclimation showed a marked increase in annual productivity of lichen- and moss-dominated biocrusts at site T1 (Fig. 7) when the seasonal acclimation of several physiological parameters was included in the model (from -27.3 to 3.1 g C m$^{-2}$ yr$^{-1}$ and from -28.6 to 15.7 g C m$^{-2}$ yr$^{-1}$).

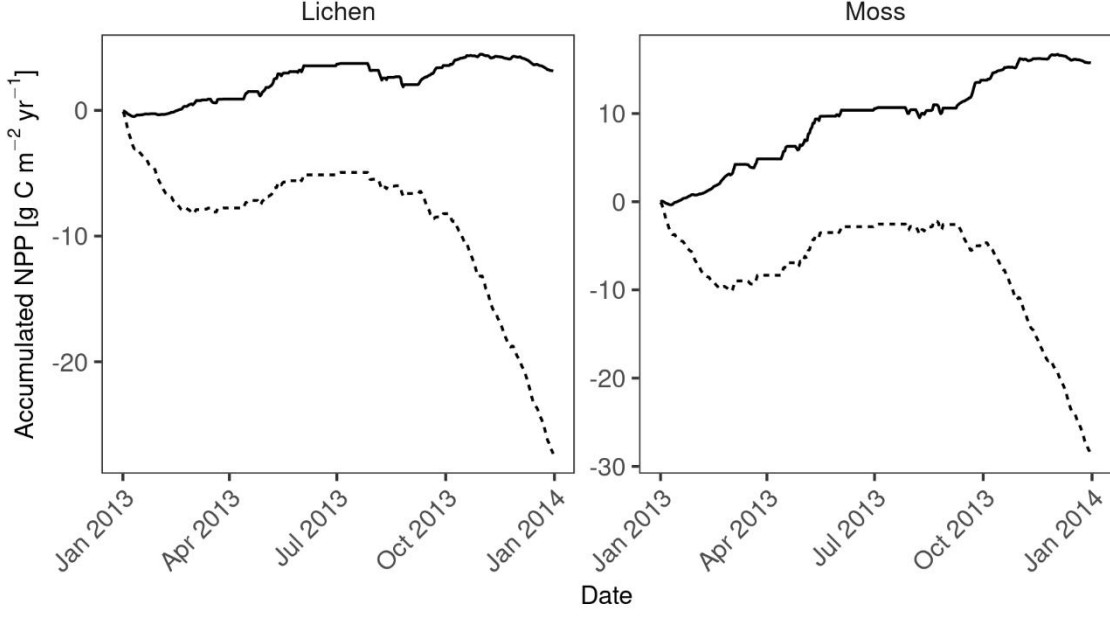

**Figure 7**: Comparison of accumulated annual C balance between simulations with dynamic parameters and fixed ones of lichens and mosses at site T1. For the simulation with the fixed parameters, all parameters that have been calibrated or measured remained constant throughout the simulation year. For the simulation with dynamic parameters at site T1, parameter metabolic respiration cost per surface area (Resp_main) was reduced by half, light absorption fraction in cells (extL) was doubled but restricted to one, the ratio of Jmax to Vcmax (jvratio) was increased by two times from September to February. The values remained the same in other months as the ones prescribed in the simulation with fixed parameters.

## 4 Discussion

### 4.1 Simulated C balance of data-driven model



The data-driven model aims to provide observation-based estimates of the carbon fluxes of non-vascular photoautotrophs
which may serve as approximation for the C balance of vegetation in biocrust-dominated ecosystems. At the two dryland
sites, the lichen- and moss-dominated biocrusts were estimated to be carbon sinks on an annual basis. As shown in the results,
lichens took up 3.8 and 0.7 g C $m^{-2}$ $yr^{-1}$, and mosses accumulated 3.2 and 6.3 g C $m^{-2}$ $yr^{-1}$ at site D1 and D2, respectively.
The estimated C balance at the two dryland sites is consistent with the magnitude of the annual C balance of different
biocrust types reported by various studies in arid habitats. Feng et al. (2014) recorded that the biocrusts composed of lichens,
mosses, and cyanobacteria of the Mu Us Desert in China took up 3.46 to 6.05 g C $m^{-2}$ $yr^{-1}$. Brostoff et al. (2005) estimated a
larger carbon gain by lichen biocrust of 11.7 g C $m^{-2}$ $yr^{-1}$ in the Mojave Desert, USA. For cyanobacteria, an annual carbon
uptake of 0.02 to 2.3 g C $m^{-2}$ was reported for deserts (Jeffries et al., 1993). The estimated C balance values in drylands
fluctuate relatively largely, but the magnitude is consistent with the simulated results by the data-driven model at D1 and D2.
For biocrust lichens growing on rock surface in a temperate grassland, Lange (2003b) measured an annual carbon gain of
21.49 g C $m^{-2}$. Additionally, several studies estimated the carbon budget in humid tundra habitats. An amount of ~12–70 g C
$m^{-2}$ $yr^{-1}$ carbon was fixed by moss-dominated biocrust, for instance (Schuur et al., 2007). The magnitude of these values
corresponds to the estimation of the C balance at T3. However, the estimated annual carbon losses of lichens and mosses by
the data-driven model in temperate regions T1 and T2 should actually lead to the death of these organisms, which is not
consistent with their dominant abundance in the field and is much lower than published by previous studies.
The mismatches of trait values between strategies selected by the LiBry model and collected species indicate that strategies
corresponding to the collected species could not maintain a positive long-term C balance, and thus were not able to survive
in the LiBry model. This is consistent with the results of the data-driven model, which also simulates a negative C balance
for those trait values. Nevertheless, there are some uncertainties in the model simulation. A bias in model estimates could
result, for instance, from missing physiological processes and trade-offs. Potential reasons for the inconsistency between
models and observations are discussed further below.

### 4.2 Potential factors influencing the C balance

Despite diverse climatic conditions, we found similarities regarding the dominant environmental factors controlling the C
balance. As shown in the results, $CO_2$ and air temperature were two most important factors at all sites that impact C balances
in the model. Relative air humidity, partly precipitation, and light intensity were also essential for C balance. The relative
importance of these factors varied slightly among climatic regions and the effects of these factors on the C balance were
different in direction among sites. We cannot rule out that the magnitudes of changes in environmental factors that we
applied in the sensitivity analysis were not balanced, which may have led to an overestimation of the relative importance of
certain factors, such as air temperature, for instance, compared to the others. The spatial patterns across climate regions of a
given environmental factor, however, are not affected by this, which means that differences between climatic regions for a
given factor are most likely robust. Hence, rainfall and $CO_2$ are likely to have the largest effect on C balance in drylands



compared to other regions, while air temperature is more relevant in alpine regions and light and relative air humidity have a
higher impact in temperate than in other regions. Nevertheless, we only studied the sensitivity of the C balance of biocrusts
dominated by the lichen *Psora decipiens* and *Cladonia furcata* (at T3), and there are variations between lichens of different
growth forms and between biocrust types. For example, cyanolichens increase in abundance with increasing rainfall, but
trebouxioid lichens have their physiological optimum in drier conditions (Phinney et al., 2021). Moreover, the impact of
precipitation on isidiate lichens is weaker than that of temperature (Phinney et al., 2021).
Our results suggest that warming can result in a large amount of carbon loss at all sites, with a particular large effect in the
alpine region. This can be explained by the overall less optimal water and temperature conditions associated with warming.
The simulated increasing respiratory costs with warming overcompensate gains in gross photosynthesis.
Ambient $CO_2$ concentration affects the gross photosynthesis rate to a large extent in the model. Although the intra-annual
change in air $CO_2$ concentration may be small in the field compared to other environmental factors, the increase of $CO_2$ in
the atmosphere in recent decades (IPCC 2021) may alter the long-term C balance substantially.
Light intensity is one of the essential factors for photosynthesis as simulated by our model. According to our climate forcing
data, the mean value of radiation maxima in January is 244 µmol m$^{-2}$ s$^{-1}$ at T1 and 245 µmol m$^{-2}$ s$^{-1}$ at the alpine site. During
wintertime in temperate and alpine regions, light intensity may be lower than the light saturation point (Fig.2 and S2).
Therefore, light intensity is a limiting factor of photosynthetic carbon assimilation in these regions, and increasing light
intensity can thus promote carbon accumulation. However, the higher light intensity can raise the surface temperature and
thus lead to more evaporation. More water loss would result in lower water saturation and activity especially in drylands,
which could limit the net photosynthesis rate. Therefore, at dryland D1, the increasing light intensity has the opposite impact
compared with other sites.
Biocrust water content is regulated by both rainfall and non-rainfall water inputs. The relative importance of different water
sources in mediating C balance varies in the model. Precipitation amount was not a key factor affecting the simulated
biocrust performance at one of the arid sites, which is consistent with another study (Baldauf et al., 2020). Our results
suggest that at the other dryland site D2, however, the precipitation amount is very closely associated with the C balance of
lichens. Moreover, we found that the effect of the amount of precipitation is small in relative humid temperate and alpine
regions. The effect of precipitation on C balance depends on the change in relative water saturation that follows from
precipitation event sizes and patterns. In some cases, decreased precipitation leading to a lower water saturation of biocrusts
may facilitate photosynthetic carbon gain via increasing the $CO_2$ diffusivity from the atmosphere into the chloroplast (Lange
et al., 1997). Nevertheless, reducing water saturation below a certain value can cause a decline in the duration of activity
(Proctor, 2001; Veste et al., 2008) which thus reduces carbon accumulation. Thus, there may be a rain threshold below which
decreasing rain may start having a negative effect on biocrust C balances. The threshold is likely species-specific as it is
associated with the water holding capacity of the organism. At arid region D1, despite the number of rainfall events being





lower, the size of many single events is still large. When rainfall input is varied in the model (decreased by half), the activity
and water saturation patterns of the studied organisms are similar to the ones with original rainfall (Fig. S5a and S5b). At the
temperate and alpine sites, although the reduction in rainfall lowered the saturation as well as the activity at many time
points (Fig. S5c and S5d), the organisms still remained active during most of the time (active 56% to 52% of the time during
one year as rainfall decreased by half) and even still fully active, since there was frequent and considerable rainfall in the
year of data collection. Thereby, the negative impact induced by reduced activity can be compensated by the positive impact
caused by reduced saturation. Thus, in the model, the impact of precipitation amount is small on C balance in these regions.
Moreover, the difference in effects of precipitation between two drylands may result from the different precipitation patterns.
The precipitation events are more evenly distributed throughout the year in D2 compared to D1, and many single events are
small in size. Decreasing the amount of precipitation alters the activity and water saturation patterns of the investigated
organisms to a larger extent in D2 as compared to D1 (Fig. S5e and f). Therefore, the amount of precipitation has a relatively
larger impact on the simulated C balance at site D2. Several studies found strong effects of variations in the precipitation
pattern on biocrust C balance (such as Reed et al., 2012). Our simulation results thus highlight the need for combined
application of field experiments and data-driven modelling to improve our understanding of differential responses to
variation in precipitation.
Furthermore, our findings indicate that relative humidity plays an important role in mediating the C balance of lichens in arid
regions, which is consistent with the results of Baldauf et al. (2020). Our results also imply that relative humidity is a crucial
factor at humid sites. However, in contrast to rainfall, the non-rainfall water inputs, such as water vapor and dew, that
depend on relative humidity have contrasting impacts on the simulated C balances of temperate, alpine, and dryland
biocrusts (Fig. 6). Whereas the increase in relative humidity can reduce the annual carbon gain in temperate regions, it
causes an increase at D1. The possible reason for this opposite direction of effect could be that the effect of air humidity
depends on the moisture conditions of the site. Higher relative air humidity could enhance water vapor and dew uptake and
reduce evaporation. At humid temperate and alpine sites, this yields larger water saturation and more periods with extremely
high saturation at sub-optimal, light-limited conditions. However, in drylands, the non-rainfall water uptake in the form of
dew or water vapor is greatest before sunrise (Chamizo et al., 2021; Ouyang et al., 2017). Especially in coastal deserts (like
the Succulent Karoo) increased fog and nocturnal dewfall in combination with higher humidity and shading mainly lead to
prolonged activated periods in the early morning when the organisms start assimilating carbon (Veste and Littmann, 2006).
Moreover, the reduced evaporation mitigates effects of drying and inactivity of organisms that occur especially at midday.
These two processes consequently result in a markedly increased annual C balance in the model.
However, the beneficial impact of the increased humidity is not common in all drylands. At site D2, our results showed an
apparent decrease in annual C balance with increased humidity. This could result from the different calculated reference
respiratory costs of the investigated organisms at these two sites from their photosynthesis in response to temperature data.
During nights with higher humidity, the surface temperature of organisms increases due to less evaporative cooling, which



increases the respiratory carbon loss at night. Moreover, higher humidity increases the activity and activates organisms that
are otherwise inactive at night (annual mean humidity at night is 66% at D1 and 70% at D2). Thus, more carbon will be lost
due to longer periods of respiration in the dark or at low light. The reference respiratory cost of the measured organisms at
D2 is much larger than at D1 (1.2 and 2.5 µmol m$^{-2}$ s$^{-1}$ at D1 and D2, respectively), so the respiration rate at D2 will be
larger than D1 under similar temperature conditions. This is supported by our results that also showed a larger yearly mean
respiration rate during the night at D2 (0.35 and 1.04 µmol m$^{-2}$ s$^{-1}$ at D1 and D2, respectively; the yearly air temperature is
12.5 and 14 °C at D1 and D2, respectively). Therefore, although more carbon is assimilated during the day due to higher
humidity in both drylands, more carbon is also lost during the night. The higher carbon loss at night at site D2 is larger than
at D1, since the organisms at D2 have a higher respiration rate than at D1. This may explain the decrease of the annual C
balance with increased air humidity at site D2 in the model.

### 4.3 Estimated negative C balance using the data-driven model

Under climate change conditions, the individual environmental factors will likely interact with each other to affect organisms
(e.g. Rillig et al., 2019). The critical role of the combination of optimal air temperature, water content, and light intensity for
the growth of biocrusts is also observed in various other studies (Büdel et al., 2018; Lange, 2003a; Lange et al., 1998b).
Overall, optimal conditions are always rare within a year, which was also described by Lange (2003b). In some cases, carbon
gains during the relatively optimal conditions may not be sufficient to compensate for losses under long-term harsh
conditions, such as autumn and winter at site T1, for instance. For this reason, the simulated C balance of mosses and lichens
in temperate humid regions was mostly negative. Given their survival under field conditions, there may be some
unconsidered mechanisms in the model that allow real biocrusts to persist under these unfavourable environmental
conditions.
Seasonal acclimation of physiological traits to the current climatic conditions may play an important role in regulating the C
balance at humid sites where the organisms are active throughout the year, such as site T1 (Fig. 7). It was observed, for
instance, that the respiration of lichens shows acclimation to seasonal changes in temperature, and the maximum $CO_2$
exchange rate of the organisms remains steady throughout the year (Lange and Green, 2005). Gauslaa (2006) found a higher
chlorophyll a/b ratio in forest lichen with increasing light. Moreover, depression in quantum efficiency in summer under
extremely dry conditions has been observed (Vivas et al., 2017). These varied physiological properties of organisms within a
year could result in different photosynthesis and respiration rates, and thus different C balances in comparison to the ones
that cannot acclimate to the seasonal climate. The missing seasonal acclimation of physiological traits may explain why the
data-driven model estimated a negative C balance for biocrusts in humid regions. Also, the LiBry model does not account for
seasonal acclimation since the strategies are assumed to have constant functional properties throughout the simulation.
Therefore, this can partly account for the mismatch of traits of selected strategies by LiBry and observations.

### 4.4 Validation of the data-driven model





The validation results of the model showed an overall good fit of daily and diurnal patterns of water content and activity (Fig.
3 and S3), and C balance at D2 (shown in Fig. 4) given the uncertainties in the data used to parameterize and evaluate the
model. This indicates that the data-driven model may be a reliable tool for C balance estimation, provided that a sufficient
amount of suitable forcing data is available.
A potential explanation for the general underestimation of activity at night and morning during several periods in a year is
the larger prescribed MWC and satmin of organisms in the model compared to those of the samples from the activity
measurements. Consequently, simulated saturation was lower, but minimal saturation for being active was higher than the
samples. Thus, the activity may have been underestimated at small water inputs such as dew and water vapor, which occur
mainly during the night and in the morning hours (Fig. S6). Moreover, underestimated activity in April and June at site T1
(Fig. S3 F(b)) may have resulted from a gap in rainfall measurements during this period. Not only rainfall amount but also
timing and frequency of rainfall events are essential for the physiological responses of biocrust communities (Belnap et al.,
2004; Coe et al., 2012; Reed et al., 2012). Therefore, although the measured annual total amount of rainfall is reasonable
(424 mm at site T1), the missing rainfall during a series of days in summer at site T1 would lead us to incorrectly predict that
the biocrusts remain inactive on these days.
Moreover, the mismatch between modelled and observed $CO_2$ assimilation rates at low or high water contents at site D2 (Fig.
4) may have partly resulted from the calibration procedure. In the calibration the simulated $CO_2$ exchange rates were higher
than measurements when the saturation exceeded the optimum saturation and hardly showed any negative values at high
saturation (Fig. S2 f). In turn, the simulated $CO_2$ exchange rates of biocrusts with an extremely low water content were zero
while the measurements showed negative values (see Fig. S2 f), pointing at a certain degree of metabolic activity in natural
conditions. Furthermore, the samples used for validation were different from the ones for calibration, which can also lead to
inaccuracies.
Additionally, the ability of the model to capture seasonality variations of C balance, which have been shown by other studies
(Büdel et al., 2018; Lange, 2003a; Zhao et al., 2016), could not be evaluated here since the monitoring of C balance in the
field and collection of samples used for photosynthesis performance measurements were conducted only during October and
early November.

### 4.5 Uncertainties of long-term C balance simulated by the data-driven model

Apart from the missing seasonal acclimation of physiological traits in the data-driven model, the estimated C balance may be
inaccurate due to potential bias in estimated relative water saturation, which partly depends on prescribed MWC, a
morphological model parameter that is obtained by measurements. We varied the MWC of lichen-dominated biocrust from
site T1 by half (+/- 50%) to examine how important uncertainty in this parameter is for the estimation of the C balance. The
outcome revealed that MWC has little effect on C balance (-25.0, -27.3, -28.3 g C m$^{-2}$ yr$^{-1}$ for reduced, original and increased
MWC). Therefore, the annual carbon estimation is robust to the uncertainties with regard to the prescribed MWC.





Furthermore, the C balance estimated by the data-driven model could be affected by a bias in calibrated physiological
parameters for organisms from photosynthesis response curves. Not all organisms forming cryptogamic covers show the
same degree of depression in net photosynthesis at high water content. For instance, among lichens, there is a wide variation
in responses of net photosynthesis to water saturation (Lange et al., 1995), also between individuals (Fig. 2 and S2).
Despite potential bias, this approach provides possibilities to predict the long-term C balance of biocrusts in the field across
various climate zones, and it enables us to analyse the mechanisms driving C balance. However, in the future, the model
needs to be calibrated with a larger number of samples collected and measured in various seasons to take the acclimation of
physiological properties into account.

## 567 5 Conclusions

While all environmental factors that were examined in our study may act as key drivers for the C balance of biocrusts, they
show distinct spatial patterns of their relative impacts. At alpine sites, air temperature is likely the most relevant factor.
Relative humidity and light may be relevant for the C balance mainly in humid temperate sites, which is not obvious. In
drylands, rainfall and also ambient $CO_2$ are found to be additional relevant factors. Furthermore, the direction of effect of
relative humidity may vary between dryland and humid sites: the higher humidity can be beneficial for the C balance in arid
regions, whereas it induces carbon loss in humid temperate and alpine sites. However, these patterns in drylands depend on
the species characteristics and microclimatic conditions of the habitat. Therefore, for the generalization of the roles of water
types in various climatic zones, a larger number of different organisms at multiple sites needs to be studied.
Our study suggests that a better, more detailed understanding of the seasonal variation of physiological traits is necessary, as
acclimation may affect the C balance substantially. The season and timing of collecting and monitoring the species should be
considered in experimental studies, especially when the characteristics of species are the basis for further analyses and
forecasts to estimate the annual carbon budget. Additionally, integration of acclimation of physiological traits in models can
improve the accuracy in C balance estimation.
Mechanistic models, as an add-on to experimental approaches, are well suited to explore the responses of the C balance of
biocrusts to separate environmental factors, and the underlying mechanisms. In turn, models need to be constrained by
measurements. As a result, we recommend combining experiments, field investigations, and modelling approaches to acquire
a comprehensive understanding from all perspectives of how biocrusts respond to climate and, potentially, future climate
change.
*Code and Data Availability Statement.* Source code of the data-driven model, LiBry modelling results, R-scripts to run the
analysis in this manuscript are available in Zenodo repository at https://doi.org/10.5281/zenodo.6971250. Field and
laboratory data are available in the corresponding publications cited in the manuscript and also from the corresponding
author upon request.



*Author Contributions.* YM and PP designed the study, BW, AK, MV and JR provided the observational data. YM did the
data processing, ran the model, YM, PP, CC, BW and MB did the data analysis and interpretation, YM wrote the manuscript
and all authors revised it.
*Conflict of Interest.* The authors have no conflict of interest to declare.
*Acknowledgments.* This research is supported by the University of Hamburg. The research in Linde is funded by
Zwillenberg-Tietz Stiftung by a grant to MV. Research in South Africa was funded by the Federal Ministry of Education and
Research (BMBF), Germany, through its BIOTA project (promotion number: 01 LC 0024A), the German Research
Foundation (Project numbers WE 2393/2-1, WE 2393/2-2) and the Max Planck Society. The research was conducted with
Northern Cape research permits (No. 22/2008 and 38/2009) and the appendant export permits and lab facilities were
provided by Burkhard Büdel at the University of Kaiserslautern and Ulrich Pöschl at the Max Planck Institute for Chemistry
in Mainz. JR acknowledges the Research Projects SCIN (PRI-PIMBDV-2011-0874) and POLAR ROCKS (PID2019-
105469RB-C21), both funded by the Spanish Ministry of Science, the possibility of obtaining part of the data and analyzing
them respectively in the frame of this research. CC acknowledges funding support provided by a NERC Standard Grant
(NE/V000764/1) and the Feodor Lynen Research fellowship from the Alexander von Humboldt foundation.

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
