# Peer review of "Exploring environmental and physiological drivers of the annual"

_Biogeosciences, 2022_

## Author Comment (AC2)

Dear Editor and Reviewers,

Thank you very much for all your time involved and for reviewers' constructive comments concerning our manuscript. These comments are all valuable and helpful for improving our manuscript. Point-by-point responses to the reviewers are listed below. We have combined the answers in one document since we refer to both reviewers in each response. The response to reviewer #2 starts on page 13.

RESPONSES TO REVIEWER #1
Manuscript ID bg-2022-179

First of all, thank you very much for your thoughtful review of our manuscript. We appreciate your positive feedback that "The results provide interesting context for understanding biocrust physiology worldwide". The comments you raised are very helpful and constructive for improving our work. All authors have carefully discussed the comments. Below, we provide detailed responses that correspond to each of your comments (highlighted in bold and italics). We hope that our responses and explanations (directly after the corresponding comment in normal font, blue and italics for new text in the revised manuscript) can fully address all points.

***Comments to the Author:** the presentation needs work for simplicity and clarity.*

Thank you for your time in reviewing such a long manuscript. The length is required since our research used two modeling approaches to study the carbon balance at six different sites, which required a comprehensive methods section, and a long discussion section for explaining the various results at different sites. However, we have tried our best to clarify and shorten the sections according to the comments. In the response, we elaborate in more detail on several specific comments.

***Abstract:** more quantitative if possible. This extends to the introduction starting especially on line 50 where the text could benefit from numeric values to help the reader understand the magnitude of C stocks and fluxes that biocrusts interact with.*

Thanks for the suggestion, we agree and will add to the abstract both the reasonable carbon balance number at D1 and also the unrealistic one at T2. The abstract has been rewritten also based on the comments of reviewer #2 (see below). Furthermore, in the revised introduction, we have added the carbon balance numbers in L50 and 51 as follows:

*An annual carbon budget of 21.49 g C m$^{-2}$ based on measured field data was reported in the study of Lange (2003b) on the crustose lichen Lecanora muralis growing on a rock surface in a temperate climate of southern Germany. Furthermore, Büdel et al. (2018) estimated an annual C balance of 1.7 g C m$^{-2}$ based on measurements on cyanobacteria-dominated biocrust in an Australian dry savannah ecosystem.*

***Line 69:** when environmental conditions are in an optimal range vascular plant would usually be favored, so what constitutes 'optimal' for biocrusts here?*

Thank you for pointing this out. 'optimal' here means an ideal situation where each environmental factor is in its optimal range for the growth of biocrusts, as determined in the laboratory measurements. The optimal conditions are species-specific. For instance, based on our measured data in D1, the optimal amount of water input for photosynthesis of biocrust dominated by *Diploschistes diacapsis* is around 0.37 mm precipitation equivalent, while for biocrust dominated by moss *Didymodon rigidulus* it is 2.55 mm precipitation equivalent. The optimal water condition for biocrusts in the field then corresponds to a rainfall pattern that leads to frequent saturation in the optimal range, and that may differ from the optimal rainfall for vascular plants. We revised this sentence as: "*While highest values of productivity in biocrusts under field conditions are achieved when the environmental factors are in the range that is optimal for the specific biocrust, it has been found that biocrusts are also able to achieve metabolic activity and thus, potential productivity, under sub-optimal conditions of temperature and light.*"

**Line 92: 'a Q10 relationship'**

Thank you for pointing this out, we will correct it accordingly.

*Regarding longwave radiation, I question somewhat the use of the ERA5 data if avoidable; were local surface temperature data available at any of the sites and if so how closely do these data align with the ERA5 data? Reading on to line 127, if surface temperature are available, avoiding ERA5 in the model would be advisable.*

Thank you very much for the suggestion. First of all, we cannot compare our measured local surface temperature data of biocrusts to the ERA5 data, since our ERA5 data set does not contain surface temperature. In the original version, we tried to explain this in L136-L141, but made it clearer in the revised manuscript, and also shifted the text upwards to the end of section 2.2.1 as follows: "*...long-wave radiation and snowfall, which were taken from ERA5 dataset (https://www.ecmwf.int/en/forecasts/datasets/reanalysis-datasets/era5). Although directly measured surface temperature data are available for all sites, we use ERA5-based down-welling long-wave radiation instead to simulate surface temperature on biocrusts. This is necessary since, in our model, calculations of photosynthesis and respiration require not only surface temperature, but also depend on water saturation of biocrusts (affecting activity). However, we do not have water saturation data available at most sites. Therefore, we instead simulate the dynamic water saturation of biocrusts based on climate, via processes such as evaporation, rainfall, and dew. The calculation of evaporation and dew automatically includes the computation of a surface temperature that emerges from solving the surface energy balance, thereby including down-welling long-wave radiation. Since the simulated surface temperature that is connected to simulated water saturation slightly deviates from the observed surface temperature (see Fig. 1 and S1), we do not directly use the observed surface temperature as input in the modeling approach, to avoid inconsistencies.*"

*Line 127 should be in the previous section.*

Thank you for the suggestion, we have changed this in the revised version.

*On line 174, something more than a visual comparison is necessary. In Fig. 1 a-d (should be b-e because the left panel should be a), the consistent early peak in the simulated temperatures should be corrected for if possible because the heat capacity that entered the model is obviously*

*incorrect. I'm not sure how this interacts with the discussion 188 if temperature was approximated. When is the temperature approximated and when was it modeled?*

Thank you for the comment. We now use RMSE to quantify the calibration in the revised manuscript and we have updated the curve fitting as follows:

We have first checked the measured surface temperature at sites that show an early peak (A1, T1, T2), and corrected the measured surface temperature data at half-hourly resolution since the data was shifted due to an improper gap-filling method we have used previously. In the raw dataset at these sites, we only have dates and not times corresponding to the data, and there are a few days where data are missing for some time points. We initially filled these gaps by shifting the data afterward, which caused an overall advance in some of the data points. Now we corrected the dataset by filling each gap with the average value of the data points at the same hour of the previous and the following day and also the previous and following 1.5-hour interval of the same day. This improves the calibration results, but notably only at site T2 (as Fig. 1 shows).

[Figure]

Figure 1: Calibration results of abiotic parameters of the data-driven model by fitting the daily (a) and diurnal (b-e) patterns of surface temperature at site T2. (b) to (e) represent the patterns of average hourly surface temperature from January to March (JFM), April to June (AMJ), July to September (JAS), and October to December (OND), respectively.

However, the early peak still exists, for instance, from January to March at site T2 (Fig. 1 (b)). This may result from the measured diurnal patterns of PAR or air temperature at 2m being inconsistent with the measured surface temperature. We compared the diurnal patterns of the measured PAR, surface temperature (Measured_Ts) and air temperature at 2m (Tair), and simulated surface temperature (Simulated_Ts) from January to March, and found the different measured climate variables might have uncertainties against each other. For instance, Tair has an earlier increase than Measured_Ts, which is unusual, and also is always lower than Measured Ts, even at night, which could partly explain the underestimated surface temperature there (Fig. 2 (a)). When we shifted the PAR data to 1 hour later, we found that the early peak of simulated surface temperature is corrected (Fig. 2 (b)).

[Figure]

Figure 2: The diurnal patterns of PAR, measured surface temperature (Measured_Ts), measured air temperature at 2m (Tair), and simulated surface temperature (Simulated_Ts) from January to March at site T2. (a): The patterns of original measurements and simulation. (b): the patterns of measured variables and simulated surface temperature when PAR was shifted to 1 hour later.

In the model, we do not calibrate the surface heat capacity, since it strongly depends on the dynamic surface water content which in turn is controlled by many biotic and abiotic factors in the model. We calibrated instead the soil heat capacity and thermal conductivity in the model. However, the soil parameters do not have a strong influence on the timing of the peak in surface temperature. In addition, a sensitivity analysis of soil thermal conductivity has been conducted at T1 to check whether the bias in calibrated boundary parameters can have a large impact on the carbon balance of biocrusts. The results showed that changing soil thermal conductivity does not prevent a negative carbon balance value in the model (change from -42.8 to -37.1 and to -50.9 gC $m^{-2}$ $yr^{-1}$, respectively, for lichen-dominated biocrust when soil thermal conductivity increased or decreased 5 times). This point will be added in the Appendix.

The statement at line 188 does not refer to the measured or simulated surface temperature, but to the ambient temperature in the laboratory when carrying out the gas exchange measurements to identify the optimum temperature of photosynthesis. This is an essential parameter for estimating the photosynthesis rate and carbon balance, but has little impact on the simulation of surface temperature on biocrusts. We have made this clearer in the revised manuscript: *"... Since the ambient temperature range that was applied in the laboratory for samples from all sites except D2 was too small to capture the optimum temperature of photosynthesis reliably, we approximated the optimum temperature from the measured data set as the average surface temperature during active periods."*

*Section 2.3.2 needs improvement also on line 204 regarding the negative photosynthesis rate. This could be a negative net C flux or the Rd parameter exceeding carbon uptake, but photosynthesis itself isn't negative.*

Thank you for the suggestion. Yes, the measured net C flux is negative, and the response curves that are shown in Fig. 2 are net photosynthesis rates, not gross photosynthesis. We have corrected this in the revised manuscript.

Moreover, we intended to re-calibrate the model to reduce the net photosynthesis rate at high water saturation by reducing the parameter minimum $CO_2$ diffusivity. The new calibration results for site T1 are shown below (Fig. 3). We improved the calibration, especially at D2. However, we can only reduce the net photosynthetic rate close to 0 at high water saturation, but it is impossible to fit the strongly negative net C flux there.

The reason are the $CO_2$ diffusion pathways implemented in our data-driven model. We assume that $CO_2$ only leaves the thallus through the same route as it enters. Furthermore, we assume that the net flux between the interior of a lichen/bryophyte and the atmosphere has the same magnitude as the flux (respiration minus photosynthesis; see Fig.4). This "steady-state" assumption is similar to vascular vegetation models and is justified by the comparably small internal space for $CO_2$ storage which prevents long-term (meaning minutes) maintenance of photosynthesis under insufficient influx of $CO_2$. These assumptions do not allow the simulation of a negative net C flux under the relatively high light of the response curve setup (see also Fig. 4 (a)). As the measured response curves show, the net C flux of one sample of lichen-dominated biocrust at high water saturation (Fig. 3 (c)) is similar to the dark respiration rate obtained from the light-response curve (Fig. 3 (a)), meaning that, for this sample, the flux of $CO_2$ out of the thallus to the atmosphere at high water saturation is likely similar in magnitude to respiration rate. In this case, the gross photosynthesis rate of the sample is likely approximately zero. But, in the model, the $CO_2$ concentration inside the thallus needs to be larger than the atmospheric $CO_2$ (400 ppm) in order to achieve a negative net flux. The relatively high $CO_2$ concentration together with the ambient light level of 400 $\mu mol/m^2/s$ in the experimental setup of the water response curve, force the modelled gross photosynthesis rate to markedly exceed zero, and therefore it is impossible to achieve a large negative net C flux with the model.

The only way to simulate a negative net C flux under light is to assume that the largest part of $CO_2$ leaves the thallus via a different route (Fig. 4 (b)). In this case, the small flux of $CO_2$ from the atmosphere into the thallus at high water saturation and further into the chloroplasts, which leads to little gross photosynthesis, is overcompensated by a much larger respiration flux that directly enters the atmosphere through a different route. However, this is highly uncertain and also a bit questionable for most lichens and bryophytes. It may be possible in a lichen if the organism has a high amount of fungal biomass located above the photobionts that contain the chloroplasts, but without detailed information on the morphology, this would represent an arbitrary parametrization. Alternatively, we would have to assume that respiration of the sample in Fig. 3 (c) is substantially higher than in Fig. 3 (a), but this seems arbitrary, too.

We have clarified this in the revised manuscript as follows from L203-208: "*However, the water responses were least well-fitted, especially at high water contents. The measured net photosynthesis response rate was negative in some cases at high water saturation, but it is not possible to reproduce this negative net photosynthesis rates with our adapted Farquhar photosynthesis model for the light and $CO_2$ conditions of the laboratory setup. Under these conditions, gross photosynthesis is larger than respiration and thus $CO_2$ is required to diffuse from the atmosphere into the thallus, not out of it. Even under low diffusivity, caused by high water saturation, there will be no net diffusion of $CO_2$ from the thallus into the ambient air assuming that inward and outward flows of $CO_2$ share the same pathway and that diffusion of $CO_2$ between atmosphere and thallus is*

*in steady-state with the flux (respiration minus gross photosynthesis). For details please see Appendix.*"

[Figure]

Figure 3: Calibration of photosynthesis parameters of the model by fitting photosynthesis response curves of moss- and lichen-dominated biocrust samples to measurements at site T1. (a): net photosynthesis rate in response to light at optimum water content and 15 °C. (b): net photosynthesis rate in response to temperature at 1200 μmol m$^{-2}$ s$^{-1}$ light and optimal water content. (c): net photosynthesis rate in response to relative water saturation at 400 μmol m$^{-2}$ s$^{-1}$ light and 15 °C.

[Figure]

Figure 4: The schematic diagram of the CO$_2$ diffusion pathways. (a): the pathway in the data-driven model, which makes it impossible to fit a strongly negative net C flux. (b): The pathway that allows

a simulation of strongly negative net C flux. Please note that the figure only shows $CO_2$ fluxes. Contrary to vascular plants, $CO_2$ and water exchange are not coupled in lichens and mosses, due to lack of stomata. The model thus calculates water fluxes independently based on the surface energy balance.

Moreover, the carbon balance estimates are affected by uncertainty in physiological parameters that were calibrated based on measurements of photosynthesis response curves. We thus conducted a sensitivity analysis of the following physiological parameters: *maintenance respiration rate (Resp_main), Q10 value of respiration (q10), the optimum temperature for gross photosynthesis (Topt), factor to relate respiration to Rubisco content (Rub_ratio), light extinction rate (ExtL), minimum saturation for activation (Sat_act0), and minimum saturation for full activation (Sat_act1)* to examine to what extent the physiological parameters can affect the carbon balance of biocrusts at all study sites. The detailed procedure and results were described in the responses to reviewer #2 below. This additional analysis will be contained in the revised manuscript.

*I'm not entirely convinced about the usefulness of section 2.5 and its description was rather meandering. That being said Fig. 6a is interesting but I wish that the normalization was done differently as a normalized value of < - 10 (for the case of air temperature) is difficult to discern.*

Thank you for this comment. In the original section 2.5, the sensitivity of the annual carbon balance of biocrusts among the study sites with regard to both abiotic factors and also seasonal acclimation of physiological parameters was described, which indeed was a bit unfocused. Based on the comments of reviewer #2, we now conducted an additional sensitivity analysis on the effects of individual physiological parameters on the carbon balance, which makes the topic of section 2.5 more consistent. In the revised manuscript, we will add a summary paragraph at the beginning of section 2.5 to have an overview of the sensitivity analyses that have been performed, and shorten the description of the normalization procedure and sensitivity analysis of acclimation (see below).

Regarding the normalization method, we normalized the annual C balance value for environmental factors between different climatic zones. The normalized C balance values are now more meaningful, and also comparable among factors and sites.

The new results of the sensitivity analysis of environmental factors are presented as follows:

[Figure]

Section 2.5 in the new version of the manuscript reads as follows:

*2.5 Sensitivity analyses*

*To investigate the role of environmental factors, physiological properties, and also seasonal acclimation for the simulated annual carbon balance of biocrusts, we conducted three sensitivity analyses using our data-driven model. With this setup, we intend to put into context the effects of environmental conditions and the uncertainty associated with the physiological properties that were used to parameterize the model. We additionally explore the impact of seasonally acclimatized*

*physiological properties on carbon assimilation at site T1, since variation between seasons represents additional uncertainty in the estimation of the carbon balance.*

*2.5.1 Effects of environmental factors*

*To investigate the role that environmental factors, namely air temperature (T_air), light intensity (Light), ambient $CO_2$ concentration ($CO_2$), and different types of water sources play in regulating the C balance of biocrusts, sensitivity analyses were conducted. The different types of water sources include rainfall (Rain) and non-rainfall water inputs such as dew and water vapor, which are determined by relative air humidity (R_hum). All the environmental factors were reduced and increased by half (+/- 50%), except for T_air and R_hum. The T_air differences varied by 5 K and R_hum by 20%. Moreover, relative humidity was constrained between 0 and 100% when the varied relative humidity exceeded this range.*

*The annual C balance for each modified environmental factor was then normalized following Eq. (1), and normalized again among different environmental factors within each climatic zone for comparing the relative importance of environmental factors:*

$$Normalized\ C\ balance = \frac{C_{ij} - C_j}{|C_j|}, \qquad\qquad (1)$$

*where Cij is the C balance of factor j under operation i, and Cj is the original C balance of factor j.*

*A positive normalized C balance demonstrates an increase in annual C balance when certain environmental factors change, and a larger magnitude of the normalized C balance number demonstrates a larger effect of this environmental factor compared to a factor with a smaller value.*

*2.5.2 Effect of physiological parameters*

*The sensitivity analysis of physiological parameters was conducted for lichen-dominated biocrust at all study sites. The original parameter values were obtained by calibration to measured net photosynthesis response curves. We then varied the values of the following physiological parameters by a consistent range for all sites: maintenance respiration rate (Resp_main), Q10 value of respiration (q10), the optimum temperature for gross photosynthesis (Topt), factor to relate respiration to Rubisco content (Rub_ratio), and light extinction rate (ExtL), minimum saturation for activation (Sat_act0), and minimum saturation for full activation (Sat_act1). Specifically, we increased or decreased Resp_main, ExtL, q10, Sat_act0 by 30%, Rub_ratio and Sat_act1 by 20%, and Topt by 5 K. These parameters are chosen since they are closely related to the response of photosynthesis and respiration to water, light, and temperature. These ranges of different parameters were determined based on the observed bounds of the photosynthetic response curves of all replicates, which have large deviations between each other at most sites as shown in Fig. 2 and Fig. S2 of the manuscript. The effects of the varied physiological parameters on the carbon balance were then normalized using the same normalization method as for the environmental factors (in Sect. 2.5.1) for comparison among parameters and climatic zones.*

*2.5.3 Effect of seasonal acclimation*

*Another sensitivity analysis was performed for site T1 to investigate the impact of seasonally acclimatized physiological properties on the carbon balance. We analyzed the lichen- and moss-dominated biocrusts at site T1 as an example, because the measured time-series of activity showed*

*that in temperate sites such as T1, the organisms were active most of the time, and thus the C balance would be more sensitive to seasonally varying properties.*

*In the analysis, rather than keeping all calibrated parameters fixed throughout the simulation period of the data-driven model, the physiological parameters metabolic respiration cost per surface area (Resp_main), light absorption fraction in cells (ExtL), and the ratio of Jmax to Vcmax (jvratio) were set to another set of values in the winter months in order to adapt to the climatic conditions, since biocrusts at sites T1 were collected in summer months. These new, "dynamic" parameters were applied in an additional simulation and the resulting carbon balance was compared to the original simulation based on the "fixed" parameters. The dynamic parameters were chosen and varied based on the literature: Respiration of lichens was found to acclimate to seasonal changes in temperature (Lange and Green, 2005). Moreover, under low light, organisms showed shade-adapted physiological characteristics with low PAR compensation and saturation points (LCP and LSP; Green and Lange, 1991). These properties can be expressed by certain parameters of the data-driven model. For instance, the respiration rate is determined by the parameter Resp_main; LCP and LSP can be affected by changing the slope of the photosynthesis-light relations through the parameter ExtL; LCP and LSP can also be modified via the parameter jvratio as it influences the value of light use efficiency at unsaturated light. .*

*Accordingly, in an hourly simulation during September and December, January, and February, the parameter Resp_main was reduced to half to lighten the respiratory cost for the samples collected at site T1. The size of ExtL was doubled to increase the slope of photosynthesis-light relations. In addition, the parameter jvratio was doubled as well to enhance the light use efficiency.*

**How does the data driven model in 2.3 differ from LiBry in 2.6 especially given that LiBry doesn't fit the observations well as described in 3.2? Was there an effort to improve LiBry given the results of the study?**

Thanks for the questions. The data-driven model was used to directly estimate carbon balance of biocrusts based on measured photosynthesis response curves, while the LiBry model was used to estimate the carbon balance of a potential biocrust community at each site that is well adapted to local climate. It was used to identify missing processes and therefore LiBry was not calibrated by measurements on purpose.

More specifically, the data-driven model aims at estimating the carbon balance of biocrusts in the field based on parameters that determine their response to environmental conditions. These parameters include hydration and physiological properties that are either measured or obtained by calibration based on measurements such as net photosynthesis response curves to light, water and temperature. In contrast, the LiBry model simulates the responses of a large number of physiologically and morphologically different strategies to environmental conditions. Each strategy is defined by a unique combination of parameter values and thus represents a group of functionally identical individuals. At the level of functional properties (traits), several similar strategies together may represent a species, thereby accounting for intra-specific trait variation. The LiBry model simulates environmental filtering of strategies for given environmental conditions to mimic natural selection and to predict trait distributions of lichen and bryophyte communities. A strategy can survive in the LiBry model if it can maintain a positive carbon balance in the long-term. Therefore, the mismatch of LiBry model simulations to observations indicates that the observed species with

its parameter combination cannot have a positive carbon balance in LiBry, which is consistent with the carbon loss of parameterized biocrusts estimated by the data-driven model in T1 and T2.

Our study can help to improve the LiBry model. We found that the missing seasonal acclimation of physiological parameters could be a source of bias in estimating the carbon balance. Therefore, in the future development of the LiBry model, seasonal acclimation of parameters should be considered to achieve more accurate predictions.

We have included these points in the Discussion and Appendix in the revised manuscript (see also answers to reviewer #2).

*Line 362: not the moisture required to give them the ability to be active?*

Thank you for the comments. Yes, moisture is crucial for the activity and $CO_2$ diffusivity of biocrusts and thus an essential factor for the carbon balance, this is also the reason why rainfall and relative humidity are environmental factors we chose. As discussed in L464-L466, the rainfall amounts of most events at temperate and alpine sites are always comparably large, thus the decrease of rainfall in the sensitivity analysis would not entirely avoid a long activity period for biocrusts. Therefore, moisture might be less relevant in these regions.

*The Fig. 7 legend could use more detail. I had to search what the "fixed" and "dynamic" parameters meant. They were detailed in section 2.5, where these terms could have been more clearly defined.*

Thank you for pointing this out. We have defined the term "fixed" and "dynamic" parameters in section 2.5.3 (see above) as names for two simulations in this sensitivity analysis. Moreover, the legend will be improved as follows:

*"Figure 7: Comparison of accumulated annual C balance between two simulations in the sensitivity analysis of seasonal acclimation of physiological properties. In the simulation "fixed parameters", all parameters that have been calibrated or measured remained constant throughout the simulation year. For the simulation "dynamic parameters" at site T1, parameter metabolic respiration cost per surface area (Resp_main) was reduced by half, light absorption fraction in cells (ExtL) was doubled but restricted to one, the ratio of Jmax to Vcmax (jvratio) was increased by two times from September to February to adapt to the winter climates. For the other months, the "fixed" values were used."*

*Line 422 and elsewhere: subscripting (here in CO_2) is inconsistent, used correctly here but not in other places.*

Thanks for pointing this out, the subscription will be checked carefully and corrected accordingly.

*For precipitation, how is dewfall and other factors that are important to biocrusts considered? In line 477 and elsewhere, is vapor pressure deficit not a more physiologically consistent approach for estimating stomatal function than relative humidity and/or is relative humidity mostly a surrogate for the surface being sufficiently wet for biocrust function to proceed?*

Thank you for the comment. In addition to rainfall, the model considers dew as water input as well via the energy balance approach (negative energy balance leads to condensation at the surface). Moreover, rainfall and dew can affect the biocrust annual carbon balance differently. For example,

the daily dew amount is usually lower than an individual rainfall event, but it is more likely in the optimal water range for the photosynthesis of biocrusts. Furthermore, dew is more frequent than rainfall events. Hence dew is likely relevant for the carbon balance of biocrusts. In addition to liquid water in the form of rainfall and dew, water vapor uptake is also a water source that can activate biocrusts (if the primary photoautotroph is not a cyanobacterium). The effects of liquid or vapor uptake on the photosynthesis of biocrusts are different (Lange, 2001).

Vapor pressure deficit is a consistent approach, and we also use vapor pressure deficit as a factor in the calculations of evaporation and dewfall in the data-driven model. Unlike vascular plants that have stomata, biocrusts are poikilohydric, and cannot actively control water exchange. Uptake (or loss) of water from (or to) the air in the model depends on the difference between saturation water vapor pressure at the surface corrected by water potential inside the biocrust thallus and the atmospheric water vapor pressure which is related to relative humidity. The relative humidity is used in the model as climate forcing data also because it is commonly monitored by weather stations at our study sites. We have clarified this in the Appendix of the revised manuscript in the "Details about the LiBry model".

*Line 569 and elsewhere: wasn't there just one alpine site such that a more accurate summary would be "at an alpine site"?*

Thank you for the suggestion, we will correct all the phrase "*at alpine sites*" to "*at an alpine site*" or "*at the alpine site*" thoroughly in the revised version of the manuscript.

*Line 570: "obvious" is subjective.*

Thanks for pointing this out, this word will be deleted.

RESPONSES TO REVIEWER #2
Manuscript ID bg-2022-179

Thank you very much for your thoughtful and comprehensive review of our paper. We appreciate your positive feedback ("I like this study and commend the authors for an ambitious undertaking"). All comments you raised are very valuable and helpful for improving our research. All authors have seriously and carefully discussed all these comments. Below we provide the detailed responses that correspond to each of your comments (highlighted in bold and italics). We hope that our responses and explanations (directly after the corresponding comment in normal font, blue and italics for new text in the revised manuscript) can fully address all your concerns.

***GENERAL COMMENTS***

***This is a paper that endeavors to simulate carbon balance in biocrusts. The approach is very nice and has high potential, but the model does fail in some cases and the authors should be more up front about this. Fortunately, in my view, the places where the model fails are interesting and can be discussed. The title and abstract should reflect that a model was constructed and tested and did not work in all cases, and the reasons should be enumerated and explored. This is done for one source of uncertainty, the environmental conditions; however, it is done inadequately for what is likely the larger source of error: the physiological parameters of the biocrusts.***

Thank you very much for the detailed review and acknowledgment of our approach. The data-driven model is used to estimate species-specific responses to environmental conditions that are governed by physiological parameters. Therefore, we totally agree that not only environmental conditions, but also physiological parameters may cause biases in estimating the carbon balance. Consequently, we now also take into account the physiological parameters as a source of error, in addition to the already previously tested impacts of seasonal change of several physiological parameters on annual carbon balance.

In the revised manuscript, we included a new section (see also our answer to reviewer #1):

*2.5.2 Effect of physiological parameters*

*The sensitivity analysis of physiological parameters was conducted for lichen-dominated biocrust at all study sites. The original parameter values were obtained by calibration to measured net photosynthesis response curves. We then varied the values of the following physiological parameters by a consistent range for all sites: maintenance respiration rate (Resp_main), Q10 value of respiration (q10), the optimum temperature for gross photosynthesis (Topt), factor to relate respiration to Rubisco content (Rub_ratio), and light extinction rate (ExtL), minimum saturation for activation (Sat_act0), and minimum saturation for full activation (Sat_act1). Specifically, we increased or decreased Resp_main, ExtL, q10, Sat_act0 by 30%, Rub_ratio and Sat_act1 by 20%, and Topt by 5 K. These parameters are chosen since they are closely related to the response of photosynthesis and respiration to water, light, and temperature. And the ranges of different parameters were determined based on the observed bounds of the photosynthetic response curves of all replicates, which have large deviations between each other at most sites as shown in Fig. 2 and Fig. S2 of the manuscript. The effects of the varied physiological parameters on the carbon balance*

*were then normalized using the same normalization method as for the environmental factors (in Sect. 2.5.1) for comparison among parameters and climatic zones.*

The results are shown in the following figure:

[Figure]

*Figure New1: (a) The effects of physiological parameters – maintenance respiration rate (Resp_main), Q10 value of respiration (q10), the optimum temperature for gross photosynthesis (Topt), factor to relate respiration to Rubisco content (Rub_ratio), and light extinction rate (ExtL), minimum saturation for activation (Sat_act0), and minimum saturation for full activation (Sat_act1) – on the annual C balance of lichens in different climate regions. The parameters decreased or increased based on the measured deviation in photosynthesis response curves of replicates. The altered annual C balance resulting from increasing or decreasing parameters is normalized by the original C balance. The colored columns indicate the average value of the normalized C balance at sites with similar climate conditions. Various styles of black points indicate different sites. (b) Relative importance of each physiological parameter compared to other parameters across the climatic regions. Larger relative importance implies a more important effect the parameter has on the C balance compared to other parameters in the given climatic region, and vice versa.*

In the Results section of the revised version of the manuscript, we will describe the results in the following way: *"We found that physiology plays an important role in all regions. In particular, the respiration-related parameters such as q10, Resp_main, and Topt have a notably higher impact on carbon balance estimation."*

**Carbon balance numbers should be listed and emphasized in the abstract, both the believable and unbelievable ones.**

Thank you for your suggestion, we will describe the carbon balance numbers at D1 and T2 in the abstract. The rewritten abstract is copied below.

**As the model does not always work, the claims about uncovering drivers and mechanisms should be substantially reduced to maybe a speculative hint here or there, not proffered as major claims in the title and abstract.**

Thank you for the suggestion. In the revised manuscript, we will make it clearer that we aim to explore the biocrust carbon balance and its drivers across different climatic zones, both environmental and physiological ones, and that we test potential underlying mechanisms, such as acclimation, rather than providing precise estimates of carbon fluxes for each climate region.

We intend to compare the relative importance of various environmental factors and physiology in the carbon balance of biocrusts among climatic zones. We think even though the data-driven model failed to estimate the carbon balance at some sites, the comparison of different sites is valid since the measurement procedure is consistent. Moreover, the patterns of relative importance remain similar when excluding the sites with strongly negative carbon balance (T1, T2, and A1). The effects of environmental factors at remaining sites are shown in the following figure (Fig. New2), and we will add it to Appendix in the revised manuscript as well:

[Figure]

*Figure New2: The effects of environmental factors - CO₂ concentration (CO₂), relative air humidity (R_hum), rainfall amount (Rain), air temperature (T_air) and light intensity (Light) on the annual C balance of lichens at different sites with reasonable carbon balance estimates.*

Therefore, the conclusions regarding the comparison of the relative importance of the environmental factors across climatic conditions may be valid. The title and abstract in the revised version of the manuscript can be found below.

***A question I am left with is: after seeing the failure at some sites to estimate a positive C balance, does this mean the dryland ones are also wrong and giving what might be a right number for the wrong reason, or does the model genuinely work better at those sites and if so why. These things are touched on but should be the main focus of the discussion.***

Thank you for this good question. We elaborated on this point in the subsection "Uncertainties of long-term C balance simulated by the data-driven model" of the discussion in the revised manuscript (also see below):

*In comparison to the unrealistic C balance numbers at T1, T2, and A1, we estimated more reasonable values in drylands and at T3. However, we do not make a definitive statement about whether or not the model predicts an accurate carbon balance in drylands, since the measured climate data and photosynthesis response curves that were used for calibrating land surface properties and various physiological parameters represent only samples of the large physiological and climatic variation. However, a higher accuracy would be more likely to be expected in drylands as these regions have a more uniform climate throughout the year regarding temperature and light levels than temperate regions that show substantial seasonality. Additionally, variation in light conditions is slightly more relevant for the simulated carbon balance than variation in moisture (see Fig. 6) because the organisms are able to become inactive, meaning that the dry season in drylands does not have a decisive effect on the carbon balance, while low light in winter in temperate climate does since organisms have to be active then. Furthermore, the longer inactive period in drylands could reduce the error in the magnitude of the simulated carbon balance caused*

*by incorrectly estimated physiological parameter values. We estimated a smaller absolute change in annual carbon balance in drylands with varied physiological parameters in the sensitivity analysis (for instance, the C balance changed by 34.6 g C m$^{-2}$ yr$^{-1}$ for parameter Topt at T1, while it changed only by 1.5 g C m$^{-2}$ yr$^{-1}$ at D1).*

**Generally, I want to emphasize again that physiological parameters being a likely source of uncertainty needs more attention above and beyond the possible effects of seasonal acclimation.**

Yes, we totally agree that the physiological parameters are likely a large source of uncertainty. We have performed sensitivity analyses to explore the role of physiological parameters in the carbon balance estimation of biocrusts at different sites under contrasting climatic conditions. The results are shown in the figure above, and we will report this in the revised version of the manuscript.

**SPECIFIC COMMENTS**

**Title: going back and re-reading the title, I think it is not accurate to say 'drivers are determined' for sites where the authors later explain that some of the C balance numbers are quite unrealistic (eg -96 g/m2/yr).**

Thank you for the suggestion. We will improve the title, accordingly, showing the role of both environmental factors and physiological uncertainties in estimating the carbon balance of biocrusts under different climatic conditions. The title in the revised manuscript reads as follows:

*Exploring environmental and physiological drivers of the annual carbon budget of biocrusts from various climatic zones with a mechanistic data-driven model*

**Abstract**

**L22. 'along a climatic gradient' is pretty vague at this point in the abstract and I am having trouble following what was done. How big is this gradient?**

Thank you for this point. The climate gradient has not been quantified, it is used to demonstrate that the six study sites are located in different climatic zones with contrasting climatic conditions, especially in rainfall amounts. The climatic gradient here is mainly a moisture input gradient from high in the snow-free season of the alpine region and temperate regions to low in arid regions. We will not use the term "*climate gradient*" but *"different climatic conditions"* in the revised manuscript. The rewritten abstract is below.

**L25. effects on what? Looks from context like carbon balance, but I had to go back to previous sentences to figure this out.**

Thank you for pointing this out. It is indeed the effect on the annual carbon balance of biocrusts. We will clarify it in the abstract of the revised manuscript (see below).

**The last sentence of the abstract indicates that the key conclusions are methodological while the title and introductory section suggests there will be new mechanistic insights. The previous sentence about climate change came as a surprise since climate change was not mentioned before that and the stated conclusion in this sentence is also vague and unsatisfying. With this mix of basic system function, applied stuff like climate change, and methodological issues, the abstract leaves the impression that the study will be unfocused. [Note: upon reading the whole**

*paper, it is more focused than I thought it would be; thus, I recommend the abstract be rewritten to reflect this.]*

Thanks for the suggestions. The conclusions related to climate change and giving methodological suggestions are based on the results of sensitivity analyses. We agree that these conclusions deviate from our main topic. Therefore, the conclusions have been revised to be more consistent with the main topic of the manuscript, which is the use of a modeling approach to estimate the carbon balance of biocrusts at sites across different climatic conditions, and the sensitivity of the carbon balance of biocrusts to environmental factors and physiological parameters. The examined sensitivity can provide insight into the potential reasons why the data-driven model succeeds or fails to estimate biocrust carbon balance under different climatic conditions. The new conclusions were rewritten in the new version of the Abstract (see below).

*Having read further in the paper, making it clear early in the abstract that this paper is mainly based on a modeling approach is recommended. I recommend to include something like "While there is a lot of empirical field data on biocrusts, rarely have these been assembled into a comprehensive modeling framework. Here we use such a framework to explore factors such as biocrust C balance in contrasting climates" I recommend to say this before talking about the environmental factors and gradients and it will make more sense to readers. Also I would back way off saying the 'key drivers are determined' based on what follows.*

Thanks for these very constructive suggestions. The data-driven model for estimating biocrust carbon balance will be highlighted at the beginning of the abstract, and we then introduce the objective of the study, which is to explore the effects of environmental factors and physiological parameters on the carbon balance of biocrusts in different climate regions. The conclusions have been rewritten in the way as stated in the response to the previous comment.

The revised Abstract is as follows:

*Biocrusts are a worldwide phenomenon, contributing substantially to ecosystem functioning. Their growth and survival depend on multiple environmental factors, including climate, and the relations of these factors to physiological processes. Responses of biocrusts to individual environmental factors have been examined in a large number of field and laboratory experiments. These observational data, however, rarely have been assembled into a comprehensive, consistent framework that allows quantitative exploration of the roles of multiple environmental factors and physiological properties for the performance of biocrusts, in particular across climatic regions. Here we used a data-driven mechanistic modeling framework to simulate the carbon balance of biocrusts, a key measure of their growth and survival. We thereby assessed the relative importance of physiological and environmental factors for the carbon balance at six study sites that differ in climatic conditions. Moreover, we examined the role of seasonal acclimation of physiological properties using our framework, since the effects of this process on the carbon balance of biocrusts are poorly constrained so far. We found substantial effects of air temperature, $CO_2$ concentration, and physiological parameters that are related to respiration on biocrust carbon balance, which differ, however, in their patterns across regions. The ambient $CO_2$ concentration is the most important factor for biocrusts from drylands while air temperature has the strongest impact at alpine and temperate sites. Maintenance respiration rate plays a more important role than optimum temperature for gross photosynthesis at the alpine site; however, this is not the case in drylands and*

*temperate regions. Moreover, we estimated a small annual carbon gain of 1.5 g m$^{-2}$ yr$^{-1}$ by lichen-dominated biocrust and 1.9 g m$^{-2}$ yr$^{-1}$ by moss-dominated biocrust at a dryland site, while the biocrusts lost a large amount of carbon at some of the temperate sites (e.g. -92.1 for lichen- and -74.7 g m$^{-2}$ yr$^{-1}$ for moss-dominated biocrust). These strongly negative values contradict the observed survival of the organisms at the sites and may be caused by the uncertainty in environmental conditions and physiological parameters that we assessed in a sensitivity analysis. Another potential explanation for this result may be the lack of acclimation in the modeling approach since the carbon balance can increase substantially when testing for seasonally varying parameters in the sensitivity analysis. We conclude that the uncertainties in air temperature, CO$_2$ concentration, respiration-related physiological parameters, and the absence of seasonal acclimation in the model for humid temperate and alpine regions may be a relevant source of error and should be taken into account in future approaches that aim at estimating the long-term biocrust carbon balance based on ecophysiological data.*

**Introduction**

**The first paragraph is an overgeneralized description of biocrusts and their function leaving me not sure where the paper is going. I recommend to hone in more clearly on setting up the modeling approach and discussion of the biocrust role in ecosystem C balance to set up the later material. The second paragraph is much better, setting up the importance of long-term C balance in biocrusts.**

Thank you for the suggestion. We designed the first paragraph to serve for setting up the importance of exploring the survival and long-term carbon balance of biocrusts in the second paragraph by describing the wide distribution of biocrusts and their important ecosystem functioning. However, we have revised the first paragraph as you recommended to more clearly show readers what our manuscript is about. We highlighted biocrusts, their importance, and different methodologies to explore them, including the cited empirical and modeling work, which is well connected to the further description of the different methodologies in the subsequent paragraph. Furthermore, we deleted the first two sentences of the second paragraph (L44 - L46) to adapt to the revised first paragraph, and the first paragraph now reads as follows:

*Non-vascular photoautotrophs, such as lichens, mosses, eukaryotic algae, and cyanobacteria, together with heterotrophic microorganisms, form biological soil crusts (biocrusts) which occur in various environments across the globe and provide a wide range of important ecosystem functions, such as build-up of soil organic carbon and nutrients (Chamizo et al., 2012; Dümig et al., 2014; Belnap et al., 2016; Ferrenberg et al., 2018). Due to the importance of biocrusts in ecosystem functioning, their growth and survival have been extensively studied, through different methodological approaches (e.g., Ladrón de Guevara et al., 2018; Lange et al., 2006; Porada et al., 2019). An established measure to quantify the growth of biocrusts is their long-term carbon balance (hereafter, C balance), which corresponds to the (accumulated) net carbon flux across the system boundaries including all relevant carbon gains and losses.*

**Finally by the end of the introduction I understand what the paper is about. It is a modeling study exploring C balance in biocrusts over a range of conditions. This needs to be MUCH more clear in the title and abstract.**

Thank you for the suggestion. We have highlighted this topic in the title and abstract as stated above.

Moreover, the last two paragraphs have been modified in the revised manuscript to adapt to the main focus that the effects of environmental factors and physiological parameters on the carbon balance of biocrusts in different climate regions, as follows:

*Most studies on the relationships between C balance and environmental factors for biocrusts are based on laboratory experiments (e.g., Coe et al., 2012; Cowan et al., 1992; Lange et al., 1998a) or direct field measurements in situ over short periods of time (e.g., Brostoff et al., 2005; Lange et al., 1994). From this work cited above, it has been recognized that the C balance of biocrusts is strongly influenced by factors such as water supply, temperature, radiation, and $CO_2$ concentration and the complex relations of these factors to physiological processes such as photosynthesis and respiration. While the highest values of productivity under field conditions are achieved when the environmental factors are in the range that is optimal for the specific biocrust, it has been found that biocrusts are also able to achieve activity and thus, potential productivity, under sub-optimal conditions of temperature and light (Colesie et al., 2016; Raggio et al., 2017, 2014). It is largely unknown, however, which relative importance each of these environmental factors and physiological parameters has for the long-term C balance of biocrusts under natural field conditions, and if the importance of factors/parameters shows a spatial and temporal pattern. In addition, seasonal acclimation of photosynthetic and respiratory properties of species to intra-annually varying climate factors found by several studies …*

*Here, we applied a mechanistic data-driven model to (a) complement empirical estimates of the annual C balance of biocrusts and (b) to address the knowledge gaps concerning the relative importance of different environmental factors and physiological parameters for the C balance along climatic gradients, thereby accounting for the role of seasonal acclimation. The advantage of this modeling approach is that it can predict at high temporal resolution the dynamic C balance of biocrust organisms for given locations by simulating the physiological processes driven by environmental factors. The model allows for a deeper mechanistic understanding of the C balance of biocrusts through factorial experiments and sensitivity analyses regarding physiological parameters and individual environmental factors …*

**2.1. I recommend that instead of making the case that the sites were chosen because they are the only sites with these data in the world (a dubious claim in my opinion - I can think of several other well-studied biocrust-focused sites that probably have enough data to take a similar approach), the authors should make the case that the sites were chosen to enhance the work done by the authors at these sites, which would be an adequate justification. The one exception to this might be the innovative 'activity measurements' the authors mention. If this is the case, I recommend to be more clear about this and explain why other proxies of activity (soil moisture perhaps) could not work at other sites.**

Thank you for the suggestions. We will elaborate on this point in the following way:

*The study sites were chosen based on data availability for carbon balance estimation, and because they cover a broad range of climatic conditions. The field and laboratory measurements conducted at all sites were following a similar protocol, which allows comparing the simulation results among sites. The necessary empirical data for C balance estimation regarding climatic conditions, species*

*physiological characteristics, and status especially in terms of moisture such as water content or activity, have been monitored in a relatively small number of experiments, so far, and the six study sites chosen here to provide a good opportunity to utilize these data for an extended modeling approach. In this context, activity measurements are more suitable than soil moisture records since they are direct, non-invasive and they do not show deviations in the temporal patterns at high resolution, which may occur with soil moisture time series.*

**L143. Soil-surface boundary layer CO2 is often higher than this due to diffusion from soil. This should be mentioned and the ramifications considered.**

Thank you for the comment. We agree that the release of $CO_2$ from soil may increase the soil-surface $CO_2$ concentration, and thus influence the C balance estimation. However, all our study sites are open sites, meaning that the $CO_2$ diffused from the soil may spread quickly. Hence, we are not sure how large the increase in surface $CO_2$ concentration is, as we did not monitor the surface $CO_2$ concentration.

We will add a sentence to mention this point in L143: "*The $CO_2$ concentration at the soil surface may be higher than 400 ppm due to the flux of respired $CO_2$ from the soil. Since our study sites are on open ground, we do not assume substantial accumulation of $CO_2$ in the near-surface boundary layer. We discuss the effect of uncertainties in $CO_2$ concentration below in section 4.2*".

Moreover, we discuss this point in the subsection "Uncertainties of long-term C balance simulated by the data-driven model" of the discussion in the revised manuscript as follows:

*As the results (Fig. 6) show, $CO_2$ concentration is an essential factor for the annual C balance of biocrusts, especially at dryland and some temperate sites. Therefore, uncertainty in the $CO_2$-value prescribed in the model may be a source of error. The $CO_2$ concentration at the surface boundary might exceed the value of 400 ppm that was prescribed in the model because of $CO_2$ diffusion from soil, which may lead to an underestimated C balance (Fig. 6a). However, with enhanced $CO_2$ concentration in the sensitivity analysis (600 ppm) at site T1, for instance, the estimated carbon balance increased only slightly, and is still strongly negative (-37.0 g C $m^{-2}$ $yr^{-1}$ for lichen and -42.4 g C $m^{-2}$ $yr^{-1}$ for moss). Hence, the lower $CO_2$ concentration can partially contribute to the strongly negative C balance at T1 and T2, but is not a major factor.*

**L144. Were these intact biocrusted soils or were the biocrusts removed from the soil and measured in an enclosed chamber separate from the soil column? Same question for L149-160.**

The $CO_2$ exchange measurements at L144 as well as L149-160 were for biocrust samples where the soil underneath the sample was removed up to the amount necessary to preserve the physical structure of the biocrusts. There may be small amounts of soil still attached to the samples at some sites, but no significant differences were found between dark respiration rates (Raggio et al, 2018), which indicates little influence of the remaining soil on gas exchange rates of biocrusts. The $CO_2$ exchange rates of clean biocrusts samples were then measured using the GFS 3000 Photosynthesis System. We have included this information in L145 in the revised version of our manuscript: "*(lichen- and moss- and also cyano-dominated biocrusts removed from surplus soil …*"; and in L152: "*The soil underneath these biocrust samples was removed up to the amount necessary to preserve the physical structure of the biocrusts*".

*L153. For poikilohydric biocrust organisms, time since hydration is a big factor in how these C balance values will look. It may be in the cited papers, but it should be discussed here too. The whole conclusions of the study could hinge on differences between, say, 1 hour vs. 4 hr vs. 24 hour wet-up periods for the biocrusts examined. Whether this has been adequately taken into account or not, it should be described how this issue was handled.*

Thank you for this important comment. Before $CO_2$-exchange measurements for biocrusts from sites were carried out, the samples were rehydrated in a climate chamber. At site T1, T2, A1 and D1, for instance, dry biocrust samples were wetted once daily for three days (Raggio et al., 2018). This is an accepted working protocol for reactivating lichens and mosses. We have added this point to L152: "*The samples were first subjected to reactivation for at least two days (D2) or three days (T1, T2, D1, A1) before measurements. At T1, T2, A1 and D1, for instance, samples were kept at 12°C under 12 h dark and 12 h light (100 μmol $m^{-2}$ $s^{-1}$) conditions for three days and wetted once a day.*"

In water content response curve measurements, samples were hydrated to the maximum water content, and then measured while drying down. The length of the activity period since hydration allowing net photosynthesis is determined in the model by the simulated water saturation that depends on the climatic forcing as well as two calibrated parameters (Sat_act0 and Sat_act1) that determine the minimum water saturation for activation and full activation, respectively.

Furthermore, regarding the potential impacts of the revitalization period or the range of water saturation that allows activity in the model on the respiration and carbon balance estimation, we included respiration-related parameters and parameters on activity estimation (Sat_act0 and Sat_act1) in the sensitivity analysis of physiological parameters. We have added this point in the revised manuscript.

*Table 1. 110 mm is pretty low. I'd probably call that arid rather than semiarid. If the determination is based on something else like aridity index, that should be reported.*

Thanks for the suggestion, we defined the climate type based solely on the annual rainfall, not the aridity index. And we will modify the type from '*semi-arid*' to '*arid*'.

*Table 2. 96 is a big loss of C. The dryland values are in line with what I would expect--small positive fluxes. These data are really valuable and this is a nice contribution of this study. Not a lot of people try to calculate these as carefully as done here. A selection of these numbers should be in the abstract to make the goals and findings of the study more concrete up front.*

Thank you for your support on the methodology and the suggestion. The specific carbon balance number at sites D1 and T2 have been demonstrated in the abstract to present examples for reasonable and unrealistic estimates. The abstract in the revised manuscript is copied above.

However, due to the comments from reviewer #1, we have reduced the parameter minimum $CO_2$ diffusivity to improve the fitting of the water-response curves at all sites and re-calibrated some physiological parameters. The estimated C balance values decreased a bit at some sites, but there are still some reasonable values in drylands and T3, and strongly negative values at T1, T2. The new results are shown in the following table (Table 2):

Table 2: Simulated annual carbon budgets of each biocrust type at all sites.

|  | Lichen | Moss | Cyanocrust |
|---|---|---|---|
|  | g C m$^{-2}$ yr$^{-1}$ | g C m$^{-2}$ yr$^{-1}$ | g C m$^{-2}$ yr$^{-1}$ |
| D1 (Almeria) | 1.5 | 1.9 |  |
| D2 (Soebatsfontein) | -1.7 | 3.1 | -8.3 |
| T1 (Gössenheim) | -42.8 | -49.7 |  |
| T2 (Öland) | -92.1 | -74.7 |  |
| T3 (Linde) | 9.4 | 18.7 |  |
| A1 (Hochtor) | -17.9 | -6.8 |  |

We also revised the corresponding descriptions in the Results and Discussion sections of the revised version of manuscript.

*Sensitivity of the environmental factors is fine and appears to be well done, but what about sensitivity to the estimates of the biocrust physiological parameters? Those are the ones that likely have much more substantial errors in my view. The high variability in these parameters among individually measured biocrusts is even noted by the authors. What if the light response or moisture curve or temp response is misshapen, have intercepts at 0 that are slightly off, etc?*

This is a crucial point, thanks for the suggestion. We performed sensitivity analyses of physiological parameters as described in the responses above. The results of the relative importance of physiology are also interpreted in the figure above (Fig. New1). Since the same gas exchange methodology has been used for sites T1, T2, A1, and D1, differences in the simulated C balance among these regions likely result from variation in the species-specific interactions between climate and physiological processes. This point is also included in the discussion section of the revised version of manuscript.

Moreover, we will add a figure (Fig. New3) in the Appendix to support the results (in the Results section) that "*even though physiology plays an important role in all regions, the C balance did not become positive when the physiological parameters were varied reasonably, that is the parameters were varied to relatively cover the deviation of response curves of replicates. Furthermore, the change in C balance value is much smaller in drylands compared to other regions.*"

[Figure]

*Figure New3: the C balance number estimated by the data-driven model without changing the parameters (Original), and with increasing and decreasing physiological parameters.*

***L398-407. Good discussion and I agree this aspect of the model throws really reasonable values, just from first principles. A shrubland that might be found in a 100-300 mm MAP ecosystem typically has an NPP on the order of 100 g/m2/yr and I would expect biocrusts to be an order of magnitude or two below that given their size, amount of chlorophyll, etc.***

Thank you, we do appreciate this acknowledgment.

***L410-413. This needs to be further unpacked. It of course makes no sense for them to lose as much carbon per year as a shrubland grows. Which part of the model is responsible for this nonsensical result?***

Thank you for pointing this out. We have explained the potential reasons for the unrealistic carbon balance numbers at these two temperate sites in section 4.3, but in the revised manuscript, we will switch the order of subsection 4.2 (The effects of environmental and physiological parameters on carbon balance; see below for the revised version) and the subsection "Uncertainties of long-term C balance simulated by the data-driven model", in which we will combine the subsection 4.3 and 4.5 and expand the explanations for the unrealistic numbers based on the uncertainties in environmental factors and physiology including our answer to other points, as follows:

[revised manuscript text omitted]

***L419. This is a key point of the paper. I recommend the authors discuss it here, not below.***

Thank you for the suggestion. In this part of the discussion, we only wanted to address the limitations and uncertainties of the LiBry model. The objective of Libry is to evaluate the negative carbon balance estimated by the data-driven model that we constructed for the study. The inconsistency of the response curves between the functional types predicted by the LiBry model for the sites and the observed species indicates that the physiological parameters that are necessary to maintain a positive carbon balance are not compatible with those of the sampled biocrusts. In other words, our measured biocrusts cannot obtain a positive carbon balance when they are represented in

the LiBry model. Since LiBry is based on the same processes as the data-driven model, this is in line with the negative values of our data-driven model simulations.

We will modify the paragraph on LiBry in the revised manuscript, while the main discussion on uncertainties of physiological parameters will be located in the new section 4.2 (see above).

"*The mismatches between strategies predicted via selection by the LiBry model for the sites and the collected species with regard to their net photosynthesis response curves indicate that the physiological parameter values that would be necessary to maintain a positive carbon balance in LiBry are not compatible with those of the sampled biocrusts. This is in line with the results of the data-driven model, which also simulates a negative C balance and is based on the same physiological processes as LiBry. This also applies to the lack of seasonal acclimation in both modeling approaches, since the strategies in LiBry are assumed to have constant functional properties throughout the simulation.*"

**L436. There are a number of field manipulations showing exactly this in Spain and USA. Could be worthwhile to cite here.**

Thank you for the suggestion, we will cite the related empirical papers in the revised manuscript (L437):

*The consistent effects of warming on C balance of biocrusts are found in various field works (e.g., Darrouzet-Nardi et al., 2015; Ladrón de Guevara et al., 2014; Li et al., 2021; Maestre et al., 2013).*

**L436-504. This section on abiotic factors is long and includes a lot of speculation. For example, the paragraph on humidity goes through a lot of hypotheticals and discussion when to me the humidity didn't stick out as a huge factor in the earlier parts of the paper, with the authors saying that co2 and air temperature were more relevant.**

Thank you for the suggestions. In fact, we found that all examined environmental factors were relevant, but $CO_2$ and air temperature were relatively more important. We discussed humidity and rainfall in such depth because we found it interesting that they have differing effects on the carbon balance, although being both related to water input, and even the same factor may have different effects on carbon balance in different climate zones. Furthermore, moisture-related factors are usually assumed to be crucial for biocrust ecophysiology, which is why we explain our findings in detail here to avoid misunderstandings. However, we will focus the discussion on the more relevant factors and findings to shorten and simplify this subsection.

Moreover, we will also add the discussion on the importance of physiological parameters to this subsection following the environmental factors. The subsection will read as follows in the revised manuscript:

[revised manuscript text omitted]

**L511. Here we get back more to what I care about: why did those estimates come out with big losses? An idea is suggested, which is that the long periods of suboptimal conditions are the problem. I would bet it goes something like this: the net C flux field/lab measurements slightly overestimate the C losses because of the timing of the respiration measurements with respect to hydration or the stresses of an incubation or a number of other factors. P lus, it's hard to separate crust from heterotrophs so you always get some heterotroph signal in those physiological measurements. Then this slightly exaggerated C loss gets multiplied by all the times**

*when the conditions are not great (most of the year) and it looks like a ton of carbon is lost. Maybe my narrative of what went wrong here is itself quite wrong, but I think if sensitivity to physiological parameters is added and then a more complete post mortem of what happened with these calculations is done, the whole study will make more sense. I want to see a story like this, but that the authors provide to the best of their ability.*

Thank you for the very helpful and reasonable suggestions. The relative importance of various environmental factors and physiological parameters has been examined. We now discuss the potential reasons leading to unrealistic carbon losses of biocrusts especially at T1 and T2 from two perspectives, namely environmental factors and physiological parameters including seasonal acclimation, in the revised manuscript in subsection 4.3 (see above).

*L515-525. Seasonal acclimation sure, but what about inaccurate estimates of physiological parameters? It's always a possibility. These things are very hard to measure. I see that the LiBry model is being used here as a talking point for why the numbers are not correct. I am not totally sure I buy the seasonal acclimation argument. It could be a factor but I think it is a lot more than that on the physiological side of things.*

Thank you for the comments. We agree that the physiological parameters, which are hard to be measured directly and are calibrated from measured photosynthesis response curves of biocrusts, are quite uncertain. If the data-driven model is sensitive to these physiological parameters, the parameters are likely to be the reason for the failure of the model simulation. We have included this point in the revised version of the manuscript (see above).

However, the seasonal variation of the physiological parameters is also relevant, since adapting a parameter to a constant value throughout the year usually does not solve problems with the C balance estimation. If, for instance, the model underestimated the physiological parameter Rub_ratio slightly, the increase of this parameter for the whole year would not make the annual carbon balance positive in the sensitivity analysis, because increased light use efficiency in winter is compensated by reduced efficiency in summer.

*L559-560. Ok yes this is what I have been saying for the whole review! The authors are aware of this. It needs to be discussed MUCH more thoroughly throughout. It doesn't make sense to exhaustively turn over all the abiotic variables when the physiology is very possibly the biggest source of error. Ideally it would be sensitivity-tested like the environmental variables. How sensitive is the model to parameter estimates on light response curve, A-Ci curve, temp response, respiration q10, etc.*

Thank you for the suggestions. We have expanded accordingly the analyses of the impacts of physiological parameters on carbon balance estimation by the data-driven model as described above. The relative importance of environmental factors will be considered as an argument for another possible source of error rather than the main findings of the study. The procedure, results, and discussion in terms of the relative importance of individual physiological parameters as well as their uncertainties that may lead to unrealistic C balance numbers are all described in the above answers.

*L563. This I agree with. It's a very nice approach, I think it just needs a more complete explanation for the modeling shortfalls. It is fine that it fails, it just has to be better described why, with quantitative information. The acclimation piece is a start, but I have a feeling the*

*shortcomings of the physiological estimates are a lot greater than just lack of accounting for acclimation. Various physiology numbers are probably slightly wrong and the model is likely sensitive to this.*

Thank you very much for the support of the approach we constructed. Yes, in addition to environmental factors and acclimation as the sources of error in carbon balance estimation, we have performed sensitivity analyses of physiological parameters to have a more comprehensive assessment of which factors or parameters are more relevant to the simulated carbon balance by the data-driven model and might thus cause the model simulation to fail.

*L569. This paper has multiple sites, but does not have an explicit spatial component so I would not use this word here.*

Thanks for the suggestion, we will consider alternative expressions: *distinct patterns of their relative impacts across climate regions*.

*L571. I do not find it particularly insightful to say rainfall is relevant in drylands. This is a given. Also I don't follow the argument about CO2 and question the assumed value used (400 ppm). I would focus the conclusion on where the model succeeds and why and vice versa.*

Thanks for the suggestions. The conclusions drawn from the results will be modified to focus more on the data-driven model estimation of biocrust carbon balance and its sensitivity to environmental factors and physiological parameters in different climatic zones.

The Conclusion section in the revised manuscript will read as follows:

*Our data-driven model provides possibilities to predict the long-term C balance of biocrusts in the field across various climate zones, and it enables us to analyze mechanisms that drive the C balance, despite marked uncertainties in the parametrization. We simulated reasonable C balance values in drylands but unrealistic ones at temperate sites with substantial seasonality. Uncertainties in environmental factors and respiration rate are likely to be the source of error for the C balance estimation since (1) all environmental factors that were examined in our study may act as relevant drivers for the C balance of biocrusts and (2) respiration-related parameters had the largest impacts compared to other physiological parameters, such as water relations or parameters solely related to Vcmax. $CO_2$ and air temperature showed the strongest effects among environmental factors and at the alpine site, the air temperature was most relevant. Compared to environmental factors, the relative impacts of physiological parameters are rather equal across climate regions. The optimum temperature may be slightly more relevant in temperate regions, while maintenance respiration rate is most important at the alpine site. Due to the importance of respiration-related physiological parameters, more studies to improve their accuracy are warranted in the future application of carbon balance modeling approaches.*

*Our study suggests that a better, more detailed understanding of the seasonal variation of physiological traits is necessary, as the more realistic estimations in drylands compared to temperate sites could be due to the weaker climate seasonality. The model needs to be calibrated with a larger number of samples collected and measured in various seasons to take the acclimation of physiological properties into account. Additionally, the integration of acclimation of physiological traits in process-based models may improve their accuracy in C balance estimation.*

*L581-585. This paragraph doesn't add much and can be removed.*

Thank you for the suggestion, we have deleted this paragraph in the revised manuscript.

Added References:

*Chamizo, S., Cantón, Y., Rodríguez-Caballero, E. and Domingo, F.: Biocrusts positively affect the soil water balance in semiarid ecosystems, Ecohydrology, 9(7), 1208–1221, doi:10.1002/eco.1719, 2016.*

*Coe, K. K., Belnap, J., Grote, E. E. and Sparks, J. P.: Physiological ecology of desert biocrust moss following 10 years exposure to elevated CO2: Evidence for enhanced photosynthetic thermotolerance, Physiol. Plant., 144(4), 346–356, doi:10.1111/j.1399-3054.2012.01566.x, 2012.*

*Darrouzet-Nardi, A., Reed, S. C., Grote, E. E. and Belnap, J.: Observations of net soil exchange of CO2 in a dryland show experimental warming increases carbon losses in biocrust soils, Biogeochemistry, 126(3), 363–378, doi:10.1007/s10533-015-0163-7, 2015.*

*Ladrón de Guevara, M., Lázaro, R., Quero, J. L., Ochoa, V., Gozalo, B., Berdugo, M., Uclés, O., Escolar, C. and Maestre, F. T.: Simulated climate change reduced the capacity of lichen-dominated biocrusts to act as carbon sinks in two semi-arid Mediterranean ecosystems, Biodivers. Conserv., 23(7), 1787–1807, doi:10.1007/s10531-014-0681-y, 2014.*

*Ladrón de Guevara, M., Gozalo, B., Raggio, J., Lafuente, A., Prieto, M. and Maestre, F. T.: Warming reduces the cover, richness and evenness of lichen-dominated biocrusts but promotes moss growth: insights from an 8yr experiment, New Phytol., 220(3), 811–823, doi:10.1111/nph.15000, 2018.*

*Lange, O.L., Reichenberger, H. and Meyer, A.: High thallus water content and photosynthetic CO2 exchange of lichens. Laboratory experiments with soil crust species from local xerothermic steppe formations in Franconia, Germany. Bot. Inst. University of Cologne, Cologne, pp.139-153. 1995*

*Lange, O. L., Allan Green, T. G., Melzer, B., Meyer, A. and Zellner, H.: Water relations and CO2 exchange of the terrestrial lichen Teloschistes capensis in the Namib fog desert: Measurements during two seasons in the field and under controlled conditions, Flora Morphol. Distrib. Funct. Ecol. Plants, 201(4), 268–280, doi:10.1016/j.flora.2005.08.003, 2006.*

*Li, X., Hui, R., Zhang, P. and Song, N.: Divergent responses of moss- and lichen-dominated biocrusts to warming and increased drought in arid desert regions, Agric. For. Meteorol., 303(March), 108387, doi:10.1016/j.agrformet.2021.108387, 2021.*

*Maestre, F. T., Escolar, C., de Guevara, M. L., Quero, J. L., Lázaro, R., Delgado-Baquerizo, M., Ochoa, V., Berdugo, M., Gozalo, B. and Gallardo, A.: Changes in biocrust cover drive carbon cycle responses to climate change in drylands, Glob. Chang. Biol., 19(12), 3835–3847, doi:10.1111/gcb.12306, 2013.*

*Porada, P., Tamm, A., Raggio, J., Cheng, Y., Kleidon, A., Pöschl, U. and Weber, B.: Global NO and HONO emissions of biological soil crusts estimated by a process-based non-vascular vegetation model, Biogeosciences, 16(9), 2003–2031, doi:10.5194/bg-16-2003-2019, 2019.*